# Common host variation drives malaria parasite fitness in healthy human red cells

Emily R Ebel[1,2], Frans A Kuypers[3], Carrie Lin[2], Dmitri A Petrov[1]*, Elizabeth S Egan[2,4]*

[1]Department of Biology, Stanford University, Stanford, United States; [2]Department of Pediatrics, Stanford University School of Medicine, Stanford, United States; [3]Children's Hospital Oakland Research Institute, Oakland, United States; [4]Department of Microbiology & Immunology, Stanford University School of Medicine, Stanford, United States

**Abstract** The replication of *Plasmodium falciparum* parasites within red blood cells (RBCs) causes severe disease in humans, especially in Africa. Deleterious alleles like hemoglobin S are well-known to confer strong resistance to malaria, but the effects of common RBC variation are largely undetermined. Here, we collected fresh blood samples from 121 healthy donors, most with African ancestry, and performed exome sequencing, detailed RBC phenotyping, and parasite fitness assays. Over one-third of healthy donors unknowingly carried alleles for G6PD deficiency or hemoglobinopathies, which were associated with characteristic RBC phenotypes. Among non-carriers alone, variation in RBC hydration, membrane deformability, and volume was strongly associated with *P. falciparum* growth rate. Common genetic variants in *PIEZO1*, *SPTA1/SPTB*, and several *P. falciparum* invasion receptors were also associated with parasite growth rate. Interestingly, we observed little or negative evidence for divergent selection on non-pathogenic RBC variation between Africans and Europeans. These findings suggest a model in which globally widespread variation in a moderate number of genes and phenotypes modulates *P. falciparum* fitness in RBCs.

*For correspondence:
dpetrov@stanford.edu (DAP);
eegan@stanford.edu (ESE)

**Competing interest:** The authors declare that no competing interests exist.

## Introduction

Malaria caused by the replication of *Plasmodium falciparum* parasites in red blood cells (RBCs) kills hundreds of thousands of children each year (*WHO, 2019*). In each 48-hr cycle of blood-stage malaria, parasites must deform RBC membranes to invade them (*Koch, 2017*; *Kariuki et al., 2020*); consume hemoglobin and tolerate the resulting oxidative stress (*Francis et al., 1997*); multiply to displace half the RBC volume (*Hanssen et al., 2012*); and remodel the RBC membrane to avoid immune detection (*Zhang, 2015*). Consequently, genetic disorders that alter aspects of RBC biology are well-known to influence malaria susceptibility (*Kwiatkowski, 2005*). For example, sickle cell trait impairs parasite growth by altering hemoglobin polymerization at low oxygen tension (*Pasvol et al., 1978*; *Archer et al., 2018*), while deficiency of the G6PD enzyme involved in oxidative stress tolerance is thought to make parasitized RBCs more susceptible to breakdown (*Ruwende and Hill, 1998*). Aside from these diseases, however, the genetic basis of RBC susceptibility to malaria remains mostly unknown.

Large genome-wide association studies (GWAS) have identified a few dozen loci that collectively explain up to 11% of the heritability of the risk of severe versus uncomplicated malaria (*Timmann et al., 2012*; *Malaria Genomic Epidemiology Network, 2014*; *Band et al., 2015*; *Leffler et al., 2017*; *Ndila et al., 2018*; *Malaria Genomic Epidemiology Network, 2019*). About 10 of the highest-confidence GWAS signals, including 6 loci known from earlier methods (*Allison, 1954*; *Field et al., 1994*; *Ruwende and Hill, 1998*; *Lell et al., 1999*; *Rowe, 2007*; *Cao and Galanello, 2010*; *Galanello and Cao, 2011*), are in or near genes expressed predominantly in RBCs. One new GWAS variant has

since been shown to regulate expression of the *ATP2B4* calcium channel (*Zámbó et al., 2017*) and to be associated with RBC dehydration (*Li et al., 2013b*), although a functional link between *ATP2B4* and *P. falciparum* replication has yet to be demonstrated. Additional GWAS discoveries of RBC variation important for malaria are not expected without massive increases in sample size (*Boyle et al., 2017*; *Malaria Genomic Epidemiology Network, 2019*), in part because of the large number of hypotheses tested. Severe malaria is a complex phenotype that combines many factors from RBCs, the vascular endothelium, the immune system, the parasite, and the environment (*Mackinnon et al., 2005*; *de Mendonça et al., 2012*). Alternate approaches are therefore needed to discover more genetic variation that impacts the replication of malaria parasites in human RBCs.

Heritable RBC phenotypes like mean cell volume (MCV), hemoglobin content (HGB/MCH), and antigenic blood type vary widely within and between human populations (*Whitfield et al., 1985*; *Evans et al., 1999*; *Pilia et al., 2006*; *Lo et al., 2011*; *Cooling, 2015*; *Canela-Xandri et al., 2018*). Large GWAS conducted mostly in Europeans have demonstrated that many blood cell phenotypes are shaped by hundreds of small-effect loci distributed throughout the genome, consistent with polygenic or omnigenic models of complex trait genetics (*van der Harst et al., 2012*; *Astle et al., 2016*; *Chami et al., 2016*; *Boyle et al., 2017*; *Chen et al., 2020*; *Vuckovic et al., 2020*). Certain blood phenotypes like average hemoglobin levels, hematocrit, and RBC membrane fragility are also known to differ between African and European populations, although the differences are typically small in magnitude (*Garn, 1981*; *Perry et al., 1992*; *Beutler and West, 2005*; *Kanias et al., 2017*; *Page et al., 2021*). This variation across populations can largely be explained by a few RBC disease alleles that have been widely selected across Africa for their protective effects on malaria (*Beutler and West, 2005*; *Lo et al., 2011*; *Kanias et al., 2017*). Despite the importance of these population-specific variants, a much larger number of common variants with small individual effects on RBC phenotypes are expected to be globally widespread (*Biddanda et al., 2020*; *Chen et al., 2020*). It remains untested whether this extensive phenotypic and genetic diversity in RBCs influences malaria susceptibility and, if so, whether it has been shaped by malaria selection.

Here, we approach these questions by performing exome sequencing and extensive RBC phenotyping on blood samples from a diverse human cohort of 122 individuals. We show that *P. falciparum* fitness varies widely among donor cells in vitro, with the distribution of parasite phenotypes in 'healthy' RBCs overlapping those from RBCs carrying classic disease alleles. We apply LASSO variable selection to identify a small set of genes and phenotypes that strongly predict parasite fitness outside of the context of RBC disease, highlighting RBC dehydration and membrane properties as key to modulating *P. falciparum* fitness. We find little evidence that non-pathogenic alleles or phenotypes that confer parasite protection are associated with African ancestry, perhaps because *P. falciparum* is susceptible to RBC variation that exists for other selective or demographic reasons. Overall, these findings advance our understanding of the origin and function of common RBC variation and suggest new targets for therapeutic intervention for malaria.

## Results
### Many healthy blood donors with African ancestry carry alleles for RBC disease

We collected blood samples from 121 donors with no known history of blood disorders, most of whom self-identified as having recent African ancestry (*Figure 1A*). As a positive control, we also sampled a patient with hereditary elliptocytosis (HE), a polygenic condition characterized by extremely fragile RBC membranes that strongly inhibit *P. falciparum* growth (*Schulman et al., 1990*; *Facer, 1995*; *Dhermy et al., 2007*; *Gallagher, 2013*). We performed whole-exome sequencing (*Figure 1—source data 1*), both to check for the presence of known RBC disease alleles and to confirm the population genetic ancestry of our donors. A principal component analysis of more than 35,000 exomic single-nucleotide polymorphisms (SNPs) showed that most donors fell along a continuum from African to European ancestry, as defined by data from the 1000 Genomes Project (*Figure 1A*). Pairwise kinship coefficients demonstrated that all donors were unrelated, apart from a six-member family with unique ancestry (*Figure 1A*, light borders). We found that 16% of the healthy donors carried pathogenic hemoglobin alleles (*Figure 1B*), including 5 heterozygotes for hemoglobin S (HbAS), 4 heterozygotes for hemoglobin C (HbAC), and 11 individuals with one or two copies of an *HBA2* deletion causing

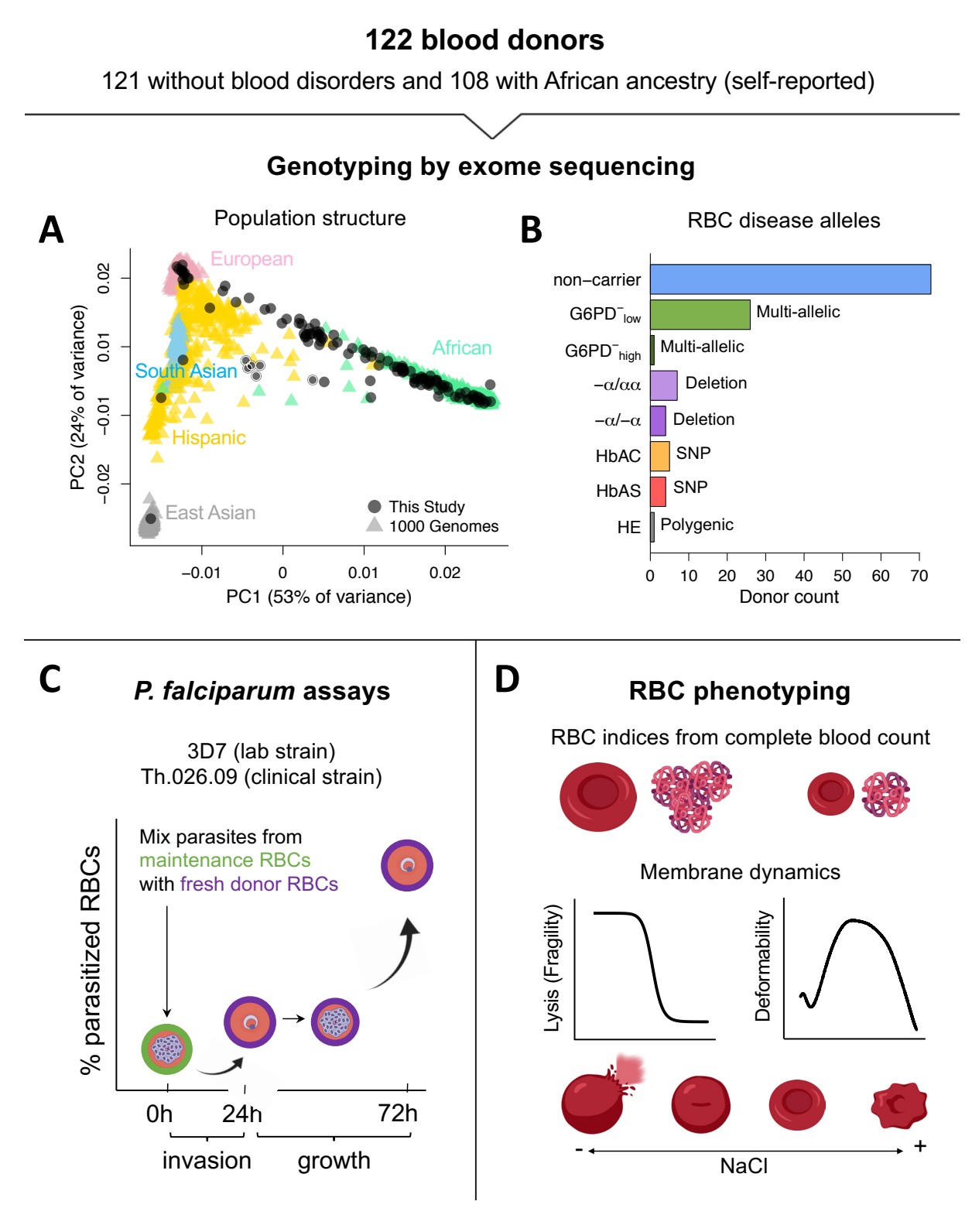

**Figure 1.** Overview of blood donors and study design. (**A**) PCA of genetic variation across 35,759 unlinked exome SNPs. Donors from this study are plotted on coordinate space derived from 1000 Genomes reference populations. Points with white borders represent six related individuals, five of whom were excluded from the study. All exome variants passing quality filters are available in *Figure 1—source data 1*. (**B**) Over a third of donors carried alleles for RBC disorders linked to *Plasmodium falciparum* resistance. Individuals with >1 disease allele were classified by their most severe

*Figure 1 continued on next page*

*Figure 1 continued*

condition. non-carrier: Donor without any of the following alleles or conditions. $G6PD^-_{low}$: Mild to medium G6PD deficiency (<42% loss of function). $G6PD^-_{high}$: Severe G6PD deficiency (>60% loss of function). $-\alpha/\alpha\alpha$: heterozygous HBA2 deletion, or alpha thalassemia minima. $-\alpha/-\alpha$: homozygous HBA2 deletion, or α-thalassemia trait. HbAC: heterozygous HBB:E7K, or hemoglobin C trait. HbAS: heterozygous HBB:E7V, or sickle cell trait. HE: hereditary elliptocytosis. (**C**) Two components of *P. falciparum* fitness were measured with flow cytometry at three timepoints. Invasion is the change in parasitemia as schizonts egress from maintenance RBCs (green) and invade fresh acceptor RBCs from the blood donors (purple). Growth is the multiplication rate from a complete parasite cycle in the fresh acceptor RBCs. (**D**) RBC phenotypes were measured using complete blood counts with RBC indices, osmotic fragility tests, and ektacytometry on fresh samples. This figure was partially created with Biorender.com. RBC, red blood cell; SNP, single-nucleotide polymorphism.

The online version of this article includes the following figure supplement(s) for figure 1:

**Source data 1.** Individual genotypes, population frequencies, and protein annotations for exome variants passing quality filters (N~160,000).

α-thalassemia (*Galanello and Cao, 2011*). We also scored eight polymorphisms in *G6PD* that have been functionally associated with various degrees of G6PD deficiency (*Yoshida et al., 1971*; *Clarke et al., 2017*) and found that 32% of the study population carried at least one, including 12 of the 20 donors with hemoglobinopathies. Among those with wild-type hemoglobin, we identified 1 individual with polymorphisms associated with severe G6PD deficiency (>60% loss of function) and 23 with polymorphisms associated with mild to medium deficiency (<42% loss of function). We detected no alleles linked to other monogenic RBC disorders, including β-thalassemia or xerocytosis (*Cao and Galanello, 2010*; *Glogowska et al., 2017*). We therefore classified the remaining 68 unrelated donors as 'non-carriers' of known disease alleles for the purposes of this work.

## *P. falciparum* replication rates vary widely among non-carrier RBCs

To determine the variation in *P. falciparum* fitness among samples with different genotypes, we performed invasion and growth assays with two parasite strains. The genome reference strain 3D7, which was originally isolated from a European, has been continuously cultured in academic labs for at least 40 years (*Walliker et al., 1987*; *Moser, 2020*). Th.026.09 is a drug-resistant strain collected from Senegal in 2009 that is minimally adapted to lab culture (*Daniels et al., 2012*). These divergent strains were selected in an attempt to balance biological realism with reliable in vitro data.

We observed a wide range of *P. falciparum* growth rates among RBC samples, especially among non-carriers that lacked known disease alleles (*Figure 2A–C*). Each strain's growth rate is defined here as parasite multiplication over a full 48-hr cycle in donor RBCs (*Figure 1C*), with the mean value for non-carriers set to 100 % after normalization. Briefly, we used a repeated control RBC sample (*Figure 2*, gray points) and other batch-specific factors to correct for variation in parasite growth across multiple experiments (*Figure 2—figure supplement 1*). Among non-carriers, growth rates ranged from 64% to 136% for 3D7 (SD=17.7%) and 76% to 128% for Th.026.09 (SD=10.6%) (*Figure 2A–B*). Per-sample growth rates were strongly correlated between the two strains (*Figure 2C*, $R^2$=0.69, $p<3×10^{-16}$) and positively correlated when measured in different weeks (p=0.35, *Figure 2—figure supplement 2*), demonstrating that these data capture meaningful variation among donor RBCs. Furthermore, as expected (*Friedman, 1978*; *Ifediba et al., 1985*; *Greene, 1993*; *Facer, 1995*), we detected reductions in mean growth rate for both strains in RBCs carrying known disease alleles. These included individuals with α-thalassemia trait (3D7 p=0.027; Th.026.09 p=0.077), HbAS (3D7 $p=1.05×10^{-7}$; Th.026.09 $p=1.2×10^{-4}$), and the single carriers of HE and severe G6PD deficiency. Notably, the wide distribution of growth rates for non-carrier RBCs had considerable overlap with the growth rates in carrier RBCs. Only the HbAS and HE samples fell entirely outside the non-carrier range. This observation implies the existence of previously unknown RBC variation that impacts *P. falciparum* growth, which may have cumulative effect sizes comparable to known disease alleles.

We observed a similarly wide range in the efficiency of *P. falciparum* invasion into donor RBCs (*Figure 2D–F*). Invasion is defined here as the fold-change in parasitemia over the first 24 hours of the assay, when parasites previously maintained in standard culture conditions egressed and invaded new donor RBCs (*Figure 1C*). Among non-carriers, invasion rates ranged from 70% to 143% for 3D7 (SD=14.9%) and 41% to 193% for Th.026.09 (SD=29.1%) (*Figure 2D–E*). Compared to growth rates, no disease alleles conferred protection against invasion that was extreme enough to fall outside the broad non-carrier range. HbAC was associated with an 11% decrease in 3D7 invasion (p=0.008), while α-thalassemia trait was associated with a 22% increase in Th.026.09 invasion (p=0.091). Only HE

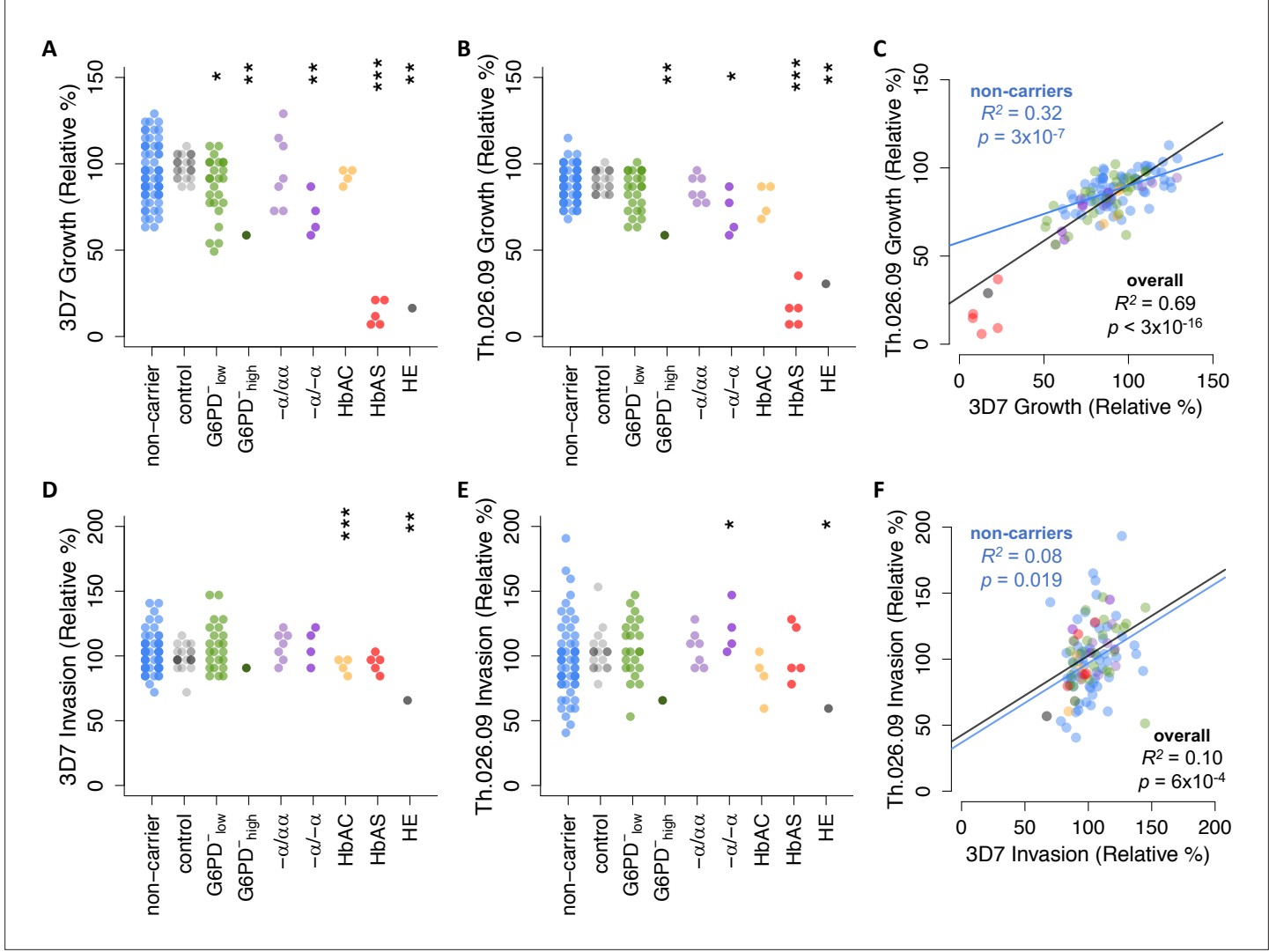

**Figure 2.** *Plasmodium falciparum* replication rate varies widely among donor RBCs. (**A, B**) Growth of *P. falciparum* lab strain 3D7 (**A**) or clinical isolate Th.026.09 (**B**) over a full 48 -hr cycle in donor RBCs (see *Figure 1C*). Growth is presented relative to the average non-carrier rate after correction for batch effects (*Figure 2—figure supplement 1*; see Materials and methods), including comparison to a repeated RBC control shown in gray. Each carrier group was compared to unrelated non-carriers using Student's t-test, except in cases where N=1, where asterisks instead indicate the percentile of the non-carrier distribution. Repeated measurements of 11 donors are shown in *Figure 2—figure supplement 2*. (**C**) Per-sample growth rates are correlated between the two *P. falciparum* strains. (**D–F**) As in (**A–C**) but for *P. falciparum* invasion efficiency (see *Figure 1C*). $R^2$ and p-values are derived from OLS regression. *p<0.1; **p<0.05; ***p<0.01. RBC, red blood cell.

The online version of this article includes the following figure supplement(s) for figure 2:

**Figure supplement 1.** Linear models of batch effects on parasite fitness.

**Figure supplement 2.** Repeatability of parasite assays in the same donors over time.

had a strong effect on the invasion efficiency of both strains. The correlation of invasion efficiencies between strains was weaker than for growth (*Figure 2F*, $R^2$=0.10, p=6×10$^{-4}$), potentially reflecting strain-specific differences in the pathways used for invasion (*Wright and Rayner, 2014*). However, we also observed greater batch effects (*Figure 2—figure supplement 1*) and greater variability between repeated samples (*Figure 2—figure supplement 2*) for invasion than for growth, suggesting that invasion is influenced by greater experimental noise.

## RBC phenotypes vary widely among non-carriers

To assess phenotypic variation across donor RBCs, we measured 22 common indices of RBC size and hemoglobin content from complete blood counts using an ADVIA hematology analyzer (*Figure 3A–D*;

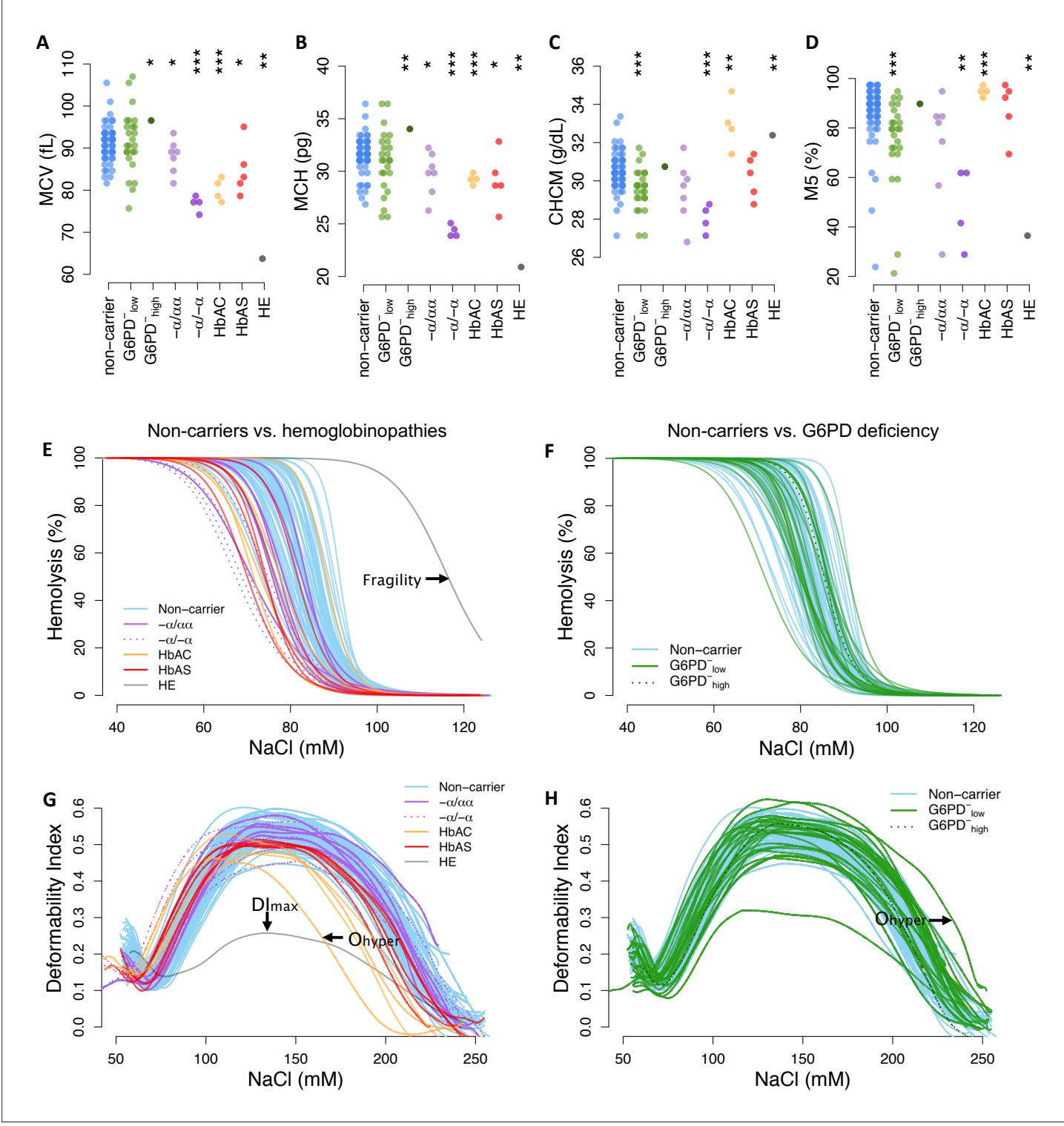

**Figure 3.** Red cell phenotypes that are abnormal in carriers also vary widely among non-carriers. (A–D) Red cell indices were measured by an ADVIA hematology analyzer. Additional indices are shown in *Figure 3—figure supplement 1*. MCV: mean corpuscular (RBC) volume; MCH: mean cellular hemoglobin; CHCM: cellular hemoglobin concentration; M5: fraction of RBCs with normal volume and normal hemoglobin (see *Figure 3—figure supplement 2*). Statistical tests as in *Figure 2*. (E, F) Osmotic fragility curves. Fragility is defined as the NaCl concentration at which 50% of RBCs lyse (see *Figure 3—figure supplement 4*). (G, H) Ektacytometry curves characterize RBC deformability and dehydration under salt stress (*Figure 3—figure supplement 5*). A heatmap of all phenotypes by carrier status is available in *Figure 3—figure supplement 3*. RBC, red blood cell.

The online version of this article includes the following figure supplement(s) for figure 3:

*Figure 3 continued on next page*

*Figure 3 continued*

**Figure supplement 1.** RBC indices data from complete blood counts in donor RBCs.

**Figure supplement 2.** RBC Matrix data from complete blood counts in donor RBCs.

**Figure supplement 3.** Heatmap of RBC phenotypes by carrier status.

**Figure supplement 4.** Osmotic fragility diagram and summary data.

**Figure supplement 5.** Ektacytometry diagram and summary data.

*Figure 3—figure supplements 1–2*). Mean cellular volume (MCV) and hemoglobin mass (MCH) are closely related traits, which can be represented together as cellular hemoglobin concentration (CHCM) or the fraction of RBCs with 'normal' hemoglobin and volume indices (M5). As expected, each known disease allele was associated with a distinct set of RBC abnormalities (*Figure 3—figure supplement 3*). These included elevated CHCM for HbAC (p=0.033), consistent with dehydration, and very low MCV (p=$6.8×10^{-5}$) and MCH (p=$2.5×10^{-7}$) for α-thalassemia trait (−α/−α), consistent with microcytic anemia (*Galanello and Cao, 2011*). RBCs from the HE patient also had very low MCV and MCH, reflecting the membrane breakage and volume loss characteristic of this disease. For all these phenotypic measures, we also observed broad distributions in non-carriers that overlapped the distributions of most carriers (*Figure 3A–D*; *Figure 3—figure supplements 1–2*). Notably, the breadth of the non-carrier distribution for each phenotype was large (e.g., 24 fl range for MCV) compared to the average difference between Africans and Europeans (e.g., 3–5 fl; *Beutler and West, 2005*; *Lo et al., 2011*). This wide diversity and substantial overlap between non-carrier and carrier traits suggest that healthy RBCs exist on the same phenotypic continuum as RBCs carrying known disease alleles.

We observed similar patterns of variation in RBC membrane fragility (*Figure 3E–F*; *Figure 3—figure supplement 4*) and membrane deformability (*Figure 3G–H*; *Figure 3—figure supplement 5*), as measured with osmotic fragility tests and osmotic gradient ektacytometry. Both sets of curves represent RBC tolerance to osmotic stress, which can result in swelling and lysis (fragility, *Figure 3E–F*) or dehydration and decreased deformability ($O_{hyper}$, *Figure 3G–H*). Specific hemoglobinopathies were associated with moderate to strong reductions in fragility, deformability, and/or resistance to loss of deformability when dehydrated (*Figure 3—figure supplements 3–5*). HE cells were both extremely fragile and extremely non-deformable. In non-carriers, the distributions for all membrane measures were wide, continuous, and overlapped the distribution of most carriers (*Figure 3E–H*). Overall, these data demonstrate that multiple phenotypic alterations associated with RBC disease alleles are also present in non-carrier RBCs.

## Non-carrier variation in RBC phenotypes predicts *P. falciparum* replication rate

To identify sets of phenotypes associated with *P. falciparum* replication in non-carrier RBCs, we used a machine learning method called LASSO (Least Absolute Shrinkage and Selection Operator) that performs regularization and variable selection (*Tibshirani, 1994*). Briefly, LASSO shrinks the regression coefficients for some possible predictors to zero to obtain a subset of predictors (in this case, phenotypes) that minimizes prediction error. This method is well-suited for data sets where predictors are correlated, as are RBC size, hemoglobin, and membrane dynamics; and for cases where the number of possible predictors is large compared to the number of measurements. To validate RBC phenotypes associated with *P. falciparum* replication by LASSO, we performed k-folds cross-validation (CV) on train and test sets derived from 10,000 divisions of the non-carrier data in 10-folds (see Materials and methods). To further control for overfitting, we also applied the same procedure to 1000 random permutations of the parasite data. Finally, for each trait selected by LASSO in at least 40% of training sets, we applied univariate OLS regression to estimate the sign of its effect on all measured components of parasite fitness. The highest-confidence results from this analysis are summarized in *Figure 4A*, with complete details provided in *Figure 4—source data 1*.

*P. falciparum* fitness in non-carrier RBCs was strongly predicted by variation in traits related to volume, hemoglobin, deformability, and dehydration (*Figure 4A*). Among 25 tested phenotypes, the most strongly predictive was the ektacytometry parameter $O_{hyper}$, which represents a cell's tendency to retain deformability in the face of dehydration (*Figure 3—figure supplement 4*). In univariate models, non-carrier RBCs with the largest $O_{hyper}$ values—that is, those that retained more deformability when

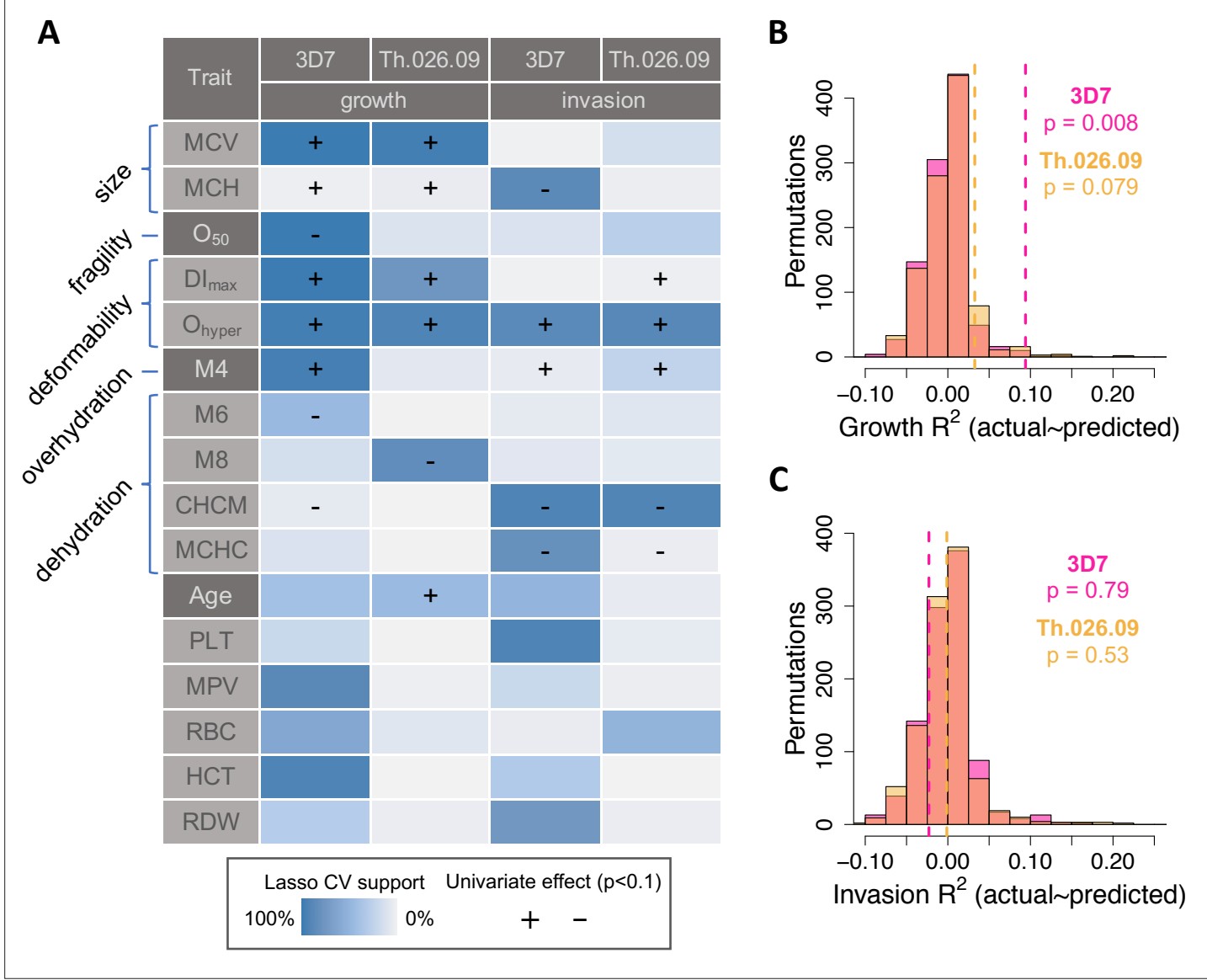

**Figure 4.** RBC phenotypes predict *Plasmodium falciparum* fitness in non-carriers. (**A**) Phenotypes selected by LASSO in at least 40% of train data sets (blue shading; see Materials and methods) in at least one of four models of parasite replication (columns). Each model was trained on ~90% of the data (**B, C**) and tested on the remaining 10% (**B, C**). (+/−) shows the direction of effect if the phenotype was significantly correlated (p<0.1) with the parasite fitness component in a separate, univariate linear model (*Figure 4—figure supplement 1*; ). MCV: mean RBC volume (fl).MCH: mean corpuscular hemoglobin (pg/RBC). $O_{50}$: Osmotic fragility (mM NaCl; see *Figure 3—figure supplement 4*). $DI_{max}$: Maximum membrane deformability (arbitrary units; see *Figure 3—figure supplement 5*). $O_{hyper}$: Tendency to resist osmotic dehydration and loss of deformability. M4: fraction of RBCs with normal volume and low hemoglobin (see *Figure 3—figure supplement 2*). M6: fraction of RBCs with normal volume and high hemoglobin. M8: fraction of RBCs with low volume and normal hemoglobin. CHCM: cellular hemoglobin concentration mean (g/dl). MCHC: mean corpuscular hemoglobin concentration (g/dl). PLT: platelet number (×$10^3$/μl). MPV: mean platelet volume (fl). RBC: red cell number (×$10^6$/μl). HCT: hematocrit, or the fraction of blood volume composed of RBCs. RDW: red cell distribution width (%). (**B, C**) Variance in parasite fitness explained by RBC phenotypes in LASSO models. Dashed lines indicate average $R^2$ for the measured test data. Each histogram shows the same procedure on 1000 permutations of the measured test data. RBC, red blood cell.

The online version of this article includes the following source data and figure supplement(s) for figure 4:

**Source data 1.** Association statistics for individual phenotypic predictors with non-zero LASSO support.

**Figure supplement 1.** Scatterplots of RBC phenotypes against parasite fitness in non-carriers.

dehydrated—supported 22–46% faster parasite growth (3D7 p=0.003, Th.026.09 p=0.007) and 31–83% more effective invasion (3D7 p=0.008, Th.026.09 p=0.005) than RBCs with the smallest $O_{hyper}$ values, which quickly lost deformability. Consistent with this result, *P. falciparum* replication was inhibited in RBCs that were more dehydrated at baseline (e.g., with higher CHCM; 3D7 invasion p=0.006, 3D7 growth p=0.024, Th.026.09 invasion p=0.004, Th.026.09 growth p=0.26). Parasites also grew faster in RBCs with larger mean volume (MCV; 3D7 p=0.001, Th.026.09 p=0.0009); a greater mass of hemoglobin per cell (MCH; 3D7 p=0.071, Th.026.09 p=0.016); and more deformable membranes ($DI_{max}$; 3D7 p=0.0005, Th.026.09 p=0.008). 3D7 growth was also reduced in RBCs with more fragile membranes ($O_{50}$; p=0.005). Additional phenotypes related to platelets and RBC density were selected for some models, but the direction of their effects was unclear when they were considered individually (*Figure 4A*, *Figure 4—source data 1*). These results indicate that common, non-pathogenic variation in RBC size, membrane dynamics, and other correlated traits have meaningful effects on *P. falciparum* replication rate in RBCs.

Taken together, the non-carrier phenotypes selected by LASSO from training data (N~61, *Figure 4A*) explained 3–9% of the variation in parasite growth in separate test data (N~7, *Figure 4B*; 3D7 p=0.008 and RMSE=18.0%; Th.026.09 p=0.079 and RMSE=10.8%). This fraction was significantly greater than expected from random permutations, which were centered on $R^2=0$ in the test data (*Figure 4B*). Notably, prediction error was greater for individuals with parasite growth values farther away from the mean. In contrast, for invasion, RBC phenotypes did not explain more variation in the test data than expected from permutations (*Figure 4C*; 3D7 p=0.79 and RMSE=15.0%; Th.026.09 p=0.53 and RMSE=29.5%). All phenotype models were less predictive for the clinical isolate Th.026.09 than the lab strain 3D7, perhaps because clinical isolates are less adapted to laboratory conditions. Overall, these results demonstrate that multiple, variable phenotypes impact *P. falciparum* susceptibility in healthy RBCs. Non-carrier cells that are less hospitable to parasites share specific traits with RBCs that carry disease alleles, including smaller size, decreased deformability, and an increased tendency to lose deformability when dehydrating.

## Common RBC alleles predict *P. falciparum* replication in non-carriers

Next, we tested whether non-carrier genotypes derived from exome sequencing could improve our predictions of *P. falciparum* replication rate. With a sample of 68 unrelated non-carriers, we lacked the power to perform the many thousands of tests that are typical in large genetic association studies (*Fadista et al., 2016*). Instead, our study design focused on 23 RBC proteins previously associated with malaria (*Figure 5—source data 1*), which we hypothesized are enriched for common variants impacting *P. falciparum* fitness, as compared to random control sets of RBC proteins. We used the same LASSO procedure described above to test 106 unlinked genetic variants (pairwise $r^2<0.1$) in these 23 RBC proteins, along with RBC phenotypes, for association with *P. falciparum* fitness in non-carriers. To test for the effects of population structure, we also included the top 10 principal components (PCs) from 1000 Genomes as possible predictors. Notably, PC1 is equivalent to the exome-wide fraction of African ancestry, as determined by ADMIXTURE with K=4 from the 1000 Genomes reference populations (see Materials and methods). We again compared these results to permuted data, as well as to 1,000 sets of 23 genes drawn at random from the RBC proteome (*Figure 5—source data 2*).

Taken together, genotypes and phenotypes selected by LASSO explained 7–15% of the variation in parasite growth rate in the test data (*Figure 5B*; 3D7 p=0.012 and RMSE=16.5%; Th.026.09 p=0.063 and RMSE=10.6%). Prediction error was greater for donors with parasite values farther away from the mean, though this trend was weaker than for phenotype-only models. The variance explained by models using real genotype and phenotype data was significantly larger than expected from permutation (*Figure 5—figure supplement 1A*) and random sets of RBC genes (*Figure 5B*), suggesting that the 23 malaria-related genes contain variation that influences *P. falciparum* development.

Nearly all of the 32 polymorphisms selected by LASSO in growth models occurred in (1) ion channel proteins, which regulate RBC hydration; (2) components of the flexible RBC membrane backbone; or (3) red cell plasma membrane proteins, including known invasion receptors (*Figure 5A*). In the first category, the highly polymorphic ion channel *PIEZO1* contained seven polymorphisms associated with small (<3.7%) to moderate (31%) reductions of *P. falciparum* growth rate. In practice, the smallest effect size that could be reliably determined for an allele with our data was ±3.7% (*Figure 5—source data 3*). The microsatellite variant *PIEZO1*-E756del, which has been a focus of several recent studies

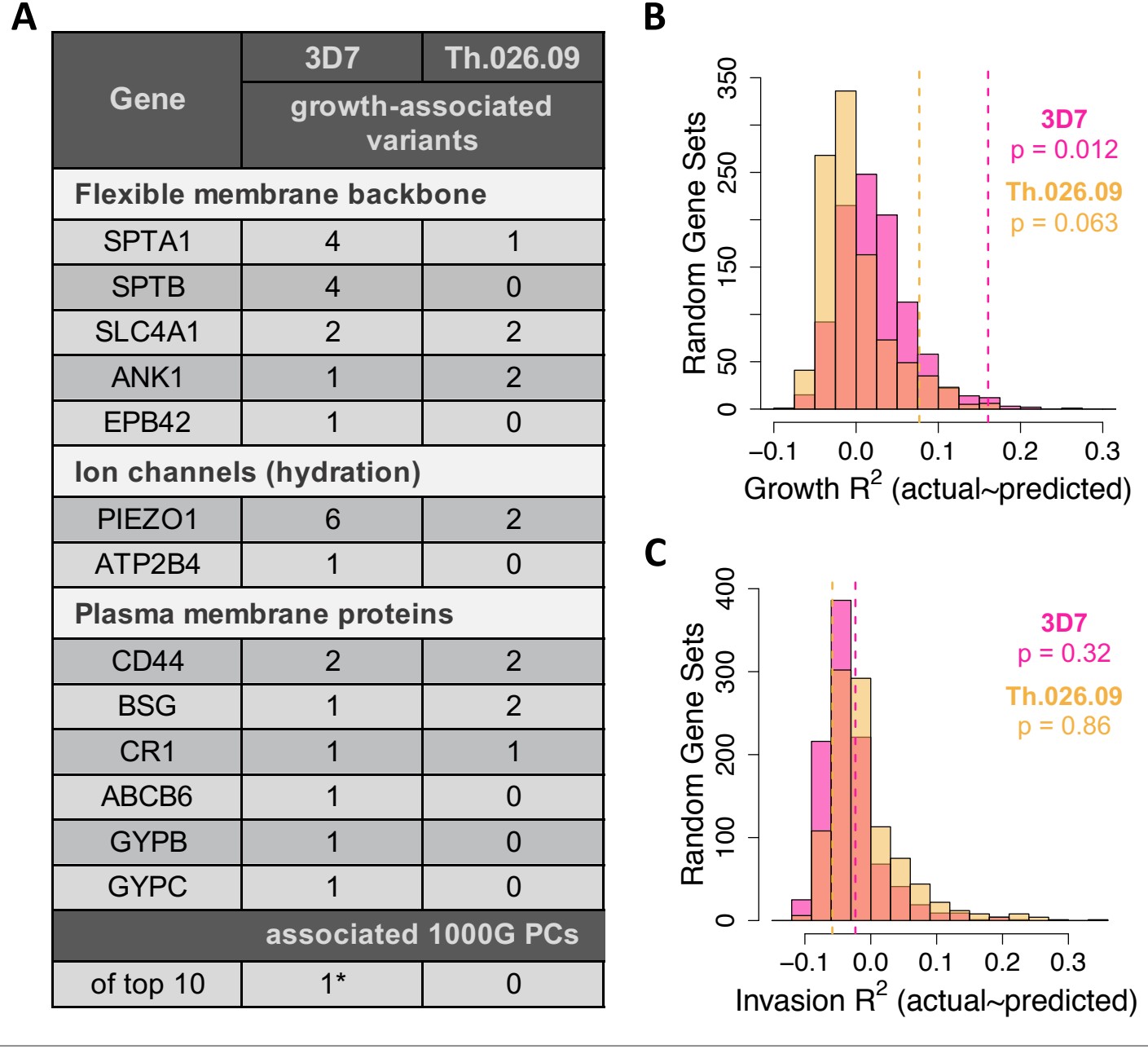

**Figure 5.** Common variation in malaria-associated genes predicts *Plasmodium falciparum* fitness in non-carrier RBCs. (**A**) Variants in 23 malaria-related genes (*Figure 5—source data 1*) and genetic PCs selected by LASSO in at least >40% of train data sets. Each model was trained on ~90% of the measured data (**B C**) and tested on the remaining 10% (**B C**). The following genes had no associated variants in non-carriers: *CD55, EPB41, FPN, G6PD, GYPA, GYPE, HBA1/2, HBB,* and *HP*. *The only significant PC association was driven by a single East Asian donor (*Figure 5—figure supplement 5*). (**B, C**) Variance in parasite fitness explained by LASSO models including 23 malaria-related genes, the top 10 PCs, and RBC phenotypes. Dashed lines indicate average $R^2$ for models using the measured test data. Each histogram shows $R^2$ for models including variants from 23 random genes in the RBC proteome (*Figure 5—source data 2*) instead of malaria-related genes. All predictors with non-zero LASSO support are shown in *Figure 5—source data 3*. Additional histograms from permuted data are shown in *Figure 5—figure supplement 1*. The variance explained by variants undiscovered by previous GWAS is shown in *Figure 5—figure supplement 4*. GWAS, genome-wide association studies; PC, principal component; RBC, red blood cell.

The online version of this article includes the following figure supplement(s) for figure 5:

**Source data 1.** Twenty-three RBC genes with strong links to malaria in the literature.

**Source data 2.** Proteins present in mature RBCs.

**Source data 3.** All genetic and phenotypic predictors with non-zero LASSO support.

*Figure 5 continued on next page*

*Figure 5 continued*

**Figure supplement 1.** Variance in parasite fitness explained by permuted data in LASSO models.

**Figure supplement 2.** Lack of association between RBC dehydration phenotypes and PIEZO1 rs59446030 or ATP2B4 rs1419114.

**Figure supplement 3.** Three non-carrier variants with potentially overdominant effects on 3D7 growth.

**Figure supplement 4.** Variants undiscovered by previous GWAS drive most of the association signal between parasite replication rate and the 23 malaria-related genes.

**Figure supplement 5.** An outlier individual for PC2 drives an apparent association between PC2 and 3D7 growth.

**Figure supplement 6.** A six-member family has unique ancestry and parasite susceptibility compared to other non-carrier donors.

(*Ilboudo et al., 2018*; *Ma et al., 2018*; *Rooks et al., 2019*; *Nguetse et al., 2020*), predicted a moderate reduction in Th.026.09 growth (–7.9%, p=0.01) but was not related to RBC dehydration in these data (*Figure 5—figure supplement 2*). For 3D7, we also detected one growth-associated variant in *ATP2B4* (–5.9%, p=0.075), which encodes the primary RBC calcium channel PMCA4b. This variant tags an *ATP2B4* haplotype implicated by GWAS in protection from severe malaria and many RBC phenotypes (*van der Harst et al., 2012*; *Li et al., 2013b*; *Lessard et al., 2017*; *Lin et al., 2020*, *Timmann et al., 2012*, *Zámbó et al., 2017*). Notably, however, this variant has never before been functionally demonstrated to be associated with *P. falciparum* fitness.

*SPTA1* and *SPTB,* which encode the flexible spectrin backbone of RBCs, contained several variants associated with the growth of at least one *P. falciparum* strain, as did the structural linker genes *ANK1*, *SLC4A1*, and *EPB42* (*Figure 5A*). We also identified a total of 10 polymorphisms in *ABCB6*, *GYPB*, *GYPC*, *CR1*, *CD44*, and basigin (*BSG*) that were associated with *P. falciparum* growth. These plasma membrane proteins have all been previously implicated in *P. falciparum* invasion by genetic deficiency studies (*Mayer et al., 2009*; *Crosnier et al., 2011*; *Egan et al., 2015*; *Egan et al., 2018*), and in some cases, studies of natural polymorphisms (*Nagayasu et al., 2001*; *Leffler et al., 2017*). Notably, two of the variants identified here are synonymous quantitative trait loci (QTL) for CD44 splicing (rs35356320) and BSG expression (rs4682) (*GTEx Consortium et al., 2017*), further supporting the possibility that they are functional. No associated variants were detected in the other 10 tested genes, including 3 hemoglobin proteins, *G6PD*, 2 glycophorins, *CD55*, *EPB41*, *FPN,* and *HP*. Taken together, these data demonstrate that dozens of host genetic variants shape the phenotypic distribution of red cell susceptibility to *P. falciparum* in non-carriers.

Eighteen of the 32 variants selected by LASSO were synonymous, which was not significantly different from the input set of 106 variants (p=0.72, two-sided binomial test). Over half of the growth-associated variants have previously been associated with gene expression traits, GWAS traits, or GWAS loci through linkage (*Figure 5—source data 3*), suggesting that they indeed tag functional polymorphisms. Novel variants nonetheless contribute substantially to the predictive power of these models (*Figure 5—figure supplement 4*), and nearly all the variants are novel in terms of association with *P. falciparum* growth rate.

In contrast to growth, models of invasion that included genotypic predictors were no more accurate than expected by chance (*Figure 5C*, p≥0.3; *Figure 5—figure supplement 1B*, p≥0.15). However, six of the nine RBC invasion receptors contained variants associated with growth (*Figure 5A*), including a SNP in glycophorin B (*GYPB*) that has been linked to malaria risk in Brazil (*Tarazona-Santos et al., 2011*). These patterns likely stem from experimental noise in our measure of invasion (*Figure 1C*; *Figure 2C and F*; *Figure 2—figure supplements 1–2*), though we note that our definition of growth involves a re-invasion event (*Figure 1C*).

No PCs of population structure were significantly associated with *P. falciparum* growth rate (*Figure 5—source data 3*), including the PC that distinguishes Africans from other populations (PC1, *Figure 1A*). One PC was selected by LASSO for 3D7 growth, but this association was driven by a single donor with East Asian ancestry and relatively high susceptibility (*Figure 5—figure supplement 5*). We note that the unique ancestry and extreme phenotypes of the six-member family (*Figure 5—figure supplement 6*) would have driven additional correlations if family members had not been excluded from the LASSO models. Although the present study is limited by sample size, these associations between global genetic PCs and *P. falciparum* growth suggest that additional functional variants remain to be discovered in many populations.

## African ancestry does not predict *P. falciparum* resistance in red cells

Based on evidence from balanced disease alleles like HbAS, it has been suggested that anti-malarial selection has shaped polygenic red cell phenotypes in African populations as a whole (*Goheen et al., 2016*; *Kanias et al., 2017*; *Ma et al., 2018*; *Page et al., 2021*). We tested this hypothesis by examining the correlation between African ancestry and *P. falciparum* fitness in non-carrier RBCs (*Figure 6A–D*). Surprisingly, we found no evidence that these traits were related, apart from a positive relationship between African ancestry and invasion rate of Th.026.09, the clinical Senegalese strain (p=0.004, $R^2$=0.13, *Figure 6D*). To understand this result, we next examined how key RBC phenotypes identified in this study (*Figure 4A*) vary with African ancestry (*Figure 6F–H*; *Figure 6—figure supplement 1*). We found that greater African ancestry predicts reduced osmotic fragility (p=1.2×10$^{-6}$), reduced RBC dehydration (CHCM p=0.009; MCHC p=0.089), and a greater fraction of 'overhydrated' RBCs with normal volume and low hemoglobin (M4 p=0.041). All of these traits actually predict *greater* red cell susceptibility to *P. falciparum* (*Figure 4A*), although together they explain less than 13% of the non-carrier variation in 3D7 growth. The remaining key phenotypes do not vary with African ancestry, which may explain why African ancestry itself is only weakly associated with *P. falciparum* fitness in non-carrier RBCs (*Figure 6A–D*).

Next, we used allele frequency data from over 54,000 individuals in the gnomAD collection (*Karczewski et al., 2020*) to test whether the polymorphisms we associated with *P. falciparum* growth occur at different frequencies in African and European populations. Geographical differences in malaria selection are sometimes hypothesized to have increased the frequency of hundreds or thousands of undiscovered anti-malarial alleles in Africa (*Mackinnon et al., 2005*; *Williams, 2006*), as has been shown for several variants causing common RBC disorders (*Kariuki and Williams, 2020*). To address this hypothesis for non-carrier variation, we calculated $F_{ST}$ between Africans and Europeans for 22 alleles with protective effects large enough to be specified in our sample (≥3.7%; *Figure 5—source data 3*). We found that 11 of these protective alleles (50%) are more common in Africans, which is not more than expected by chance (p=0.5, one-sided binomial test). The three protective variants with the largest absolute $F_{ST}$ values are all more common in Europeans, including a synonymous *SPTA1* allele with GWAS associations to several RBC and white blood cell traits. Two protective *PIEZO1* variants are more common in Africans, including E756del and a synonymous variant of large effect. Overall, however, we find no evidence that African populations are enriched for non-pathogenic RBC polymorphisms or phenotypes associated with impaired *P. falciparum* growth in vitro.

## Discussion

Healthy RBCs harbor extensive phenotypic and proteomic variation, both within and between human populations. In this study, we demonstrate that this variation modulates a wide range of RBC susceptibility to *P. falciparum* parasites. Our findings add to a growing understanding of the genetic and phenotypic basis of RBC resistance to *P. falciparum*, especially for RBCs that lack population-specific disease alleles. These findings suggest new targets for future malaria interventions, in addition to challenging assumptions about the role of malaria selection in shaping human RBC diversity.

Exponential replication of *P. falciparum* is a significant driver of malaria disease progression (*Bejon et al., 2007*). Therefore, the ample variation that we observed in this trait in vitro could be relevant for clinical outcomes in endemic regions. Growth inhibition from HbAS, for example, reduces the risk of death from malaria by reducing parasite density in the blood (*Allison, 1954*; *Luzzatto, 2012*). While HbAS has a uniquely extreme effect size, we found a threefold range of parasite replication rates among non-carrier RBCs that share substantial overlap with RBCs carrying other protective variants. Although the physiologically complex basis of severe malaria (*Okwa, 2012*) makes it difficult to estimate the precise contribution of RBC factors to severe malaria risk, the genotypes and phenotypes we have associated with *P. falciparum* fitness may contribute to malaria susceptibility.

We have shown here that widespread, 'normal' variation in RBC hydration and deformability traits are associated with *P. falciparum* fitness in non-carrier RBCs. Interestingly, the protective phenotypes we detect in non-carrier RBCs are also present in carriers, albeit to a stronger degree (*Clark et al., 1983*; *Mockenhaupt, 2000*; *Pengon et al., 2018*). These results are consistent with experimental manipulations that reduce *P. falciparum* growth, such as chemical or genetic dehydration of RBCs (*Tiffert et al., 2005*; *Ma et al., 2018*). They are also consistent with the protective effect conferred

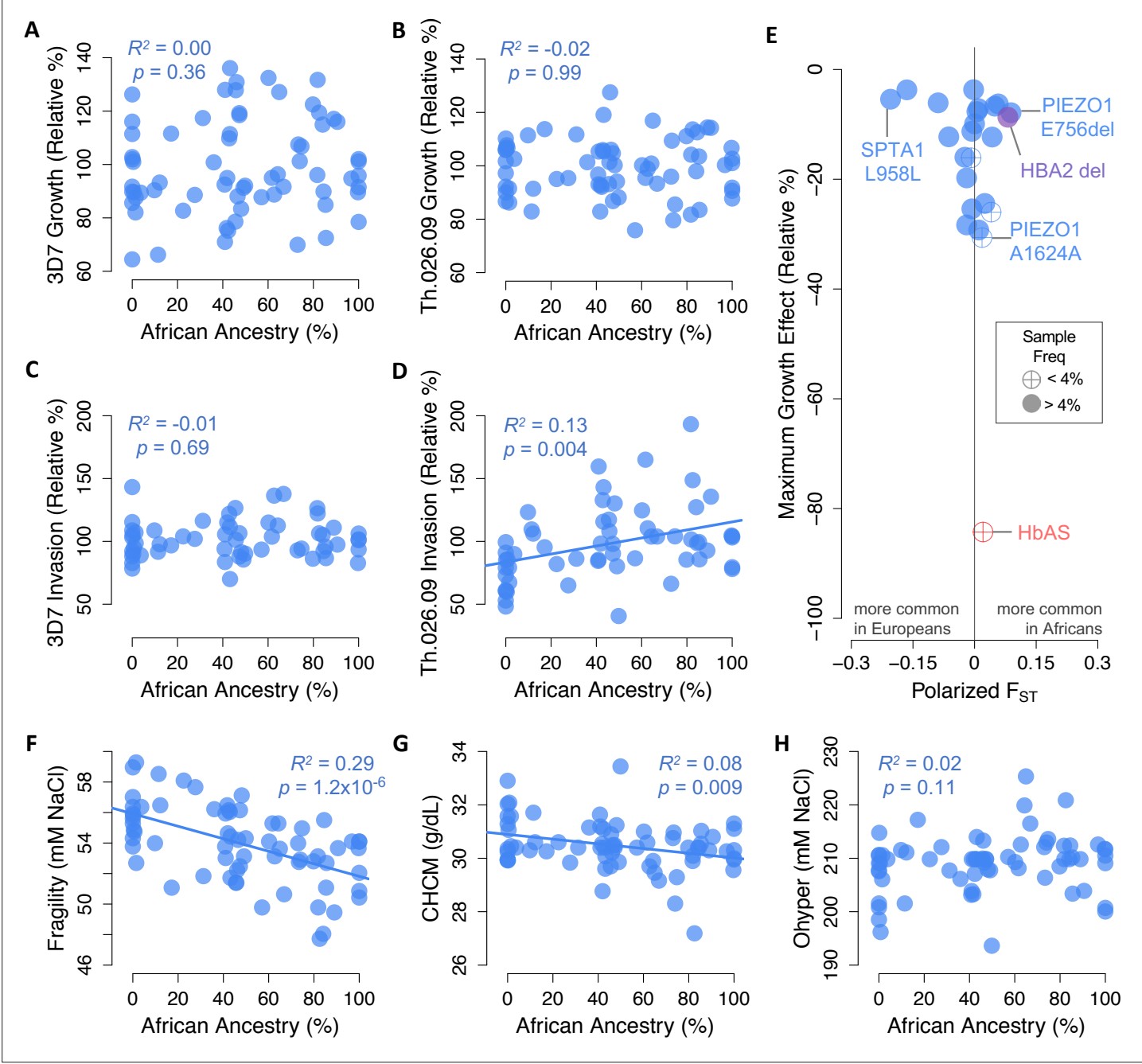

**Figure 6.** Little evidence of widespread selection in Africa for slower *Plasmodium falciparum* replication, protective alleles, or protective phenotypes in non-carriers. (**A–D**) Parasite replication versus the exome-wide fraction of African ancestry in non-carriers, determined with ADMIXTURE by comparison to 1000 Genomes reference populations. $R^2$ and p-values are shown for OLS regression. (**E**) Alleles in 23 malaria-related genes that predict slower *P. falciparum* growth in non-carriers (**Figure 5A**) are not enriched for higher frequencies in Africa versus Europe. Effect sizes are shown for one allele copy for 3D7 or Th.026.09 growth, whichever was greater. Effect sizes were determined from additive models except for three alleles that appeared overdominant (**Figure 5—figure supplement 3**). $F_{ST}$ was calculated from African and European samples in gnomAD (see Materials and methods). HbAS and the HBA2 deletion are shown for comparison. (**F–H**) RBC phenotypes associated with *P. falciparum* growth versus the exome-wide fraction of African ancestry in non-carriers. Slower *P. falciparum* growth in RBCs is predicted by greater fragility (**F**), greater dehydration (**G**), and lower $O_{hyper}$ (**H**) (**Figure 4A**). Additional phenotypes are shown in **Figure 6—figure supplement 1**.

The online version of this article includes the following figure supplement(s) for figure 6:

**Figure supplement 1.** Scatterplots of RBC phenotypes versus African ancestry in non-carriers.

by Dantu, a rare glycophorin variant associated with increased membrane tension (*Field et al., 1994*; *Leffler et al., 2017*; *Kariuki et al., 2020*). Our data expand upon these prior findings by demonstrating for the first time that common, healthy phenotypic variation in RBC traits contributes meaningfully to *P. falciparum* growth.

In the last decade, several association studies have explored the genetic basis of common variation in RBC traits using large, mostly European cohorts (*van der Harst et al., 2012*; *Astle et al., 2016*; *Chami et al., 2016*; *Canela-Xandri et al., 2018*; *Chen et al., 2020*; *Vuckovic et al., 2020*). These studies agree that the broad distribution of RBC phenotypes in humans is shaped by a large number of common alleles, similar to other complex traits (*Boyle et al., 2017*). Although the effects of most individual alleles are likely too small to be considered pathogenic on their own, different combinations of alleles may underlie the broad phenotypic variation observed in non-carriers. We cannot rule out the possibility that some extreme phenotypes could be better explained by the presence of large-effect 'disease' alleles that remain undiscovered. In particular, our study was not powered to detect rare alleles, which could be an important source of missing heritability (*Génin, 2020*; *Kierczak, 2021*). Some RBC phenotypes are also shaped by environmental variation, such as diet and time of day (*England et al., 1976*; *Sennels et al., 2011*), which likely diminishes correlations between repeated samples. Although this study cannot distinguish among these explanations for phenotypic variation among non-carrier RBCs, it does suggest that this broad variation is both healthy and functional.

In our linear models of *P. falciparum* growth, phenotypic variation among RBCs was outperformed by genetic variation in a small number of RBC proteins. This result implies the existence of protective RBC phenotypes that we did not measure (or did not measure with sufficient accuracy), such as quantitative proteomic, transcriptomic, and metabolomic traits that could be addressed by future studies. Approximately half of the polymorphisms we identified are non-synonymous and may therefore exert direct effects on phenotypes like RBC membrane structure or ion transport. The other half of associated polymorphisms were synonymous, which could be linked to coding variants but could also have direct effects on mRNA transcription, splicing, and stability (*Sauna and Kimchi-Sarfaty, 2011*). Indeed, silent and coding SNPs are equally likely to be associated with human disease (*Chen et al., 2010*), and many synonymous sites experience strong selection (*Supek et al., 2014*; *Machado et al., 2020*). Synonymous SNPs that impact splicing, like rs35356320 in CD44, may also impact protein structure. Some other conceivable RBC phenotypes, such as the dynamics of membrane modification during *P. falciparum* development, may only become evident in more detailed time course experiments. The true number of RBC phenotypes that impact *P. falciparum* may be effectively infinite (*Kinsler et al., 2020*), making it useful in practice that genetic variation is more predictive of parasite growth.

One reason that our study could identify genetic associations with a modest sample size was because we focused on a relatively well-defined component of a larger disease that lends itself to controlled, in vitro experiments. Another important explanation is our use of LASSO variable selection on a restricted set of polymorphisms in genes with strong existing links to malaria (*Flynn et al., 2017*). Focusing our tests on a limited number of hypotheses obviated the need to meet an exome-wide significance threshold, while still allowing for the discovery of novel alleles. This approach relies directly on prior knowledge (*Figure 5—source data 1*) and cannot readily be expanded to explicitly test large numbers of anonymous genetic variants. However, testing fewer hypotheses that are more likely to be true helps ensure that 'significant' results are reliable (*Ioannidis, 2005*). Exome-wide data were still critical in this study for assessing population structure, as well as for performing permutation tests that confirmed an enrichment of signal in our 23 focal genes. However, future studies with many more than 68 non-carriers will be required to discover additional associations in unknown genes, non-genic regulatory variation, and any alleles with smaller effects. It is also important to note that genetic linkage complicates the identification of the exact functional polymorphisms in any population sample (*Sohail et al., 2019*); as in GWAS, we cannot rule out that some associated variants are merely linked to the true functional variants. Indeed, about half of our associated variants occur in linkage blocks containing other SNPs associated with RBC traits by GWAS. In this way, our evidence most strongly supports the conclusion that 13 specific RBC genes are strongly enriched for polymorphisms with impacts on *P. falciparum* growth.

The associations we observed for parasite growth were stronger and more significant than the associations for parasite invasion. While batch effects clearly played a role, this may also be due to missing invasion data in 10 non-carrier samples (see Materials and methods) that reduced statistical

power. Both technical and biological reasons may drive the relatively greater noise observed in our invasion data. For example, invasion success may depend on the length of time spent outside the incubator during assay set-up as well as the genotypes of both donor and acceptor red cells. The reproducibility of our invasion data is also constrained by low and variable starting parasitemia and a 24hr time point, which could be substantially improved in future studies using live-cell imaging focused on invasion. Despite these limitations, our 'growth' measurement includes a 're-invasion' event and our growth and invasion measurements are correlated. RBC deformability and dehydration are associated with both fitness components, and SNPs in several canonical invasion receptors are only associated with growth. The invasion data also allow us to highlight unique and interesting trends in the Senegalese clinical strain and in carriers of hemoglobin C.

We also observed weaker associations for the clinical strain Th.026.09 than for the lab strain 3D7. These strains display large differences in absolute growth rate, possibly because Th.026.09 carries costly alleles for drug resistance and 3D7 has had decades longer to adapt to lab conditions (*Walliker et al., 1987*; *Daniels et al., 2012*; *Moser, 2020*). Interestingly, African ancestry predicted higher invasion only for Th.026.09, which might indicate that this strain is better adapted to African RBCs. Despite these differences, we showed that normalized fitness values were significantly correlated between the two strains across donors. Several RBC phenotypes and genotypes that predicted fitness in one strain were also replicated in the other. These results suggest that our findings may be generalizable across divergent strains of *P. falciparum*, although future studies would benefit by testing many more lab strains and clinical isolates.

One of the unique aspects of our study is the participation of individuals with a range of African ancestry, defined by similarity to donors from five 1000 Genomes reference populations. We found that African ancestry was unexpectedly associated with RBC phenotypes that *improved* parasite fitness, particularly for Th.026.09. In the future, it would be very interesting to test for local parasite adaptation to human RBCs using *P. falciparum* strains and RBC samples from around the globe. We also found that the total set of polymorphisms associated with *P. falciparum* growth by LASSO are not enriched in African populations included in the gnomAD database of human variation. Notably, a recent test of data from a large GWAS for severe malaria (*Malaria Genomic Epidemiology Network, 2019*) was also unable to demonstrate that natural selection has driven many malaria-protective alleles to higher frequencies in African versus European populations. Therefore, for the total set of alleles detectable in this study, we offer at least four possible explanations for this unexpected result. First, compared to large-effect disease alleles, the majority of non-pathogenic variants may not have had sufficient time to increase in frequency since *P. falciparum* began expanding in humans some 5000–10,000 years ago (*Sundararaman et al., 2016*; *Otto et al., 2018*). Second, the complexity of severe malaria could mean that the variants discovered here do not substantially impact disease outcome, especially relative to known disease variants. Third, the variants discovered here may have pleiotropic effects on other phenotypes, which are themselves subject to other selective pressures besides malaria resistance. Finally, human adaptation may be too local to detect with coarse-grain sampling of sub-Saharan African genetic diversity (e.g., *Pankratov et al., 2020*). Overall, however, our data suggest that few RBC alleles remain to be discovered that are both particularly common in Africa and have large effects on *P. falciparum* proliferation in RBCs.

More broadly, these data show that it may be inaccurate to make assumptions about RBC susceptibility to *P. falciparum* based on a person's race or continental ancestry. These kinds of hypotheses (*Williams, 2006*; *Goheen et al., 2016*; *Kanias et al., 2017*; *Ma et al., 2018*) are based on well-known examples of balanced disease alleles, which are notable exceptions to the overwhelming genetic similarity of all human populations (*Rosenberg et al., 2002*; *Novembre and Di Rienzo, 2009*). In our data, RBC variation that is associated with reduced *P. falciparum* fitness is clearly not limited to individuals with recent African ancestry. This result is an important reminder that >90% of the total genetic variation among humans occurs within populations, rather than across them (*Lewontin, 1972*; *Rosenberg, 2011*); and that the majority of common genetic variation is shared among all human populations (*Biddanda et al., 2020*).

In conclusion, this study demonstrates that substantial phenotypic and genetic diversity in healthy human RBCs impacts the replication of malaria parasites. Whether or not this diversity is shaped by malaria selection, a better understanding of how *P. falciparum* biology is impacted by natural RBC variation could help lead to new therapies for one of humanity's most important infectious diseases.

# Materials and methods

**Key resources table**

| Reagent type (species) or resource | Designation | Source or reference | Identifiers | Additional information |
|---|---|---|---|---|
| Biological sample (*Homo sapiens*) | Primary whole blood samples | This paper | | Freshly drawn from de-identified human subjects into CPDA tubes (IRB #40479) |
| Strain, strain background (*Plasmodium falciparum*) | 3D7 | PMID:3299700; Obtained from Walter and Eliza Hall Institute, Melbourne, Australia | | |
| Strain, strain background (*P. falciparum*) | Th026.09 | PMID:22430961; Gift from Daouda Ndiaye and Sarah Volkman, Senegal | | |
| Commercial assay or kit | DNeasy Blood and Tissue Kit | QIAGEN | | |
| Commercial assay or kit | KAPA Hyperplus Kit | Roche | | |
| Commercial assay or kit | SeqCap EZ Prime Exome Kit | Roche | | |
| Sequence-based reagent | Primers amplifying *PIEZO1* exon 17 | PMID:32265284 | | |
| Software, algorithm | bwa mem | http://arxiv.org/abs/1303.3997 | 0.7.17-r1188 | |
| Software, algorithm | GATK | https://gatk.broadinstitute.org/hc/en-us | 4.0.0.0 | |
| Software, algorithm | vcftools | doi:10.1093/bioinformatics/btr330 | 0.1.15 | |
| Software, algorithm | ANNOVAR | PMID:20601685 | 2018-04-16 | |
| Software, algorithm | PLINK | PMID:17701901 | v1.90b6.8 64-bit | |
| Software, algorithm | ADMIXTURE | PMID:21682921 | 1.3.0 | |
| Software, algorithm | R | https://www.R-project.org/ | 3.5.1 | |
| Other | SYBR Green I nucleic acid stain | Invitrogen | S7563 | |
| Other | Drabkin's Reagent | Ricca Chemical | 2660–32 | |

## Sample collection and preparation

One-hundred and twenty-one subjects with no known history of RBC disorders were recruited to donate blood at the Stanford Clinical and Translational Research Unit. This study size was designed to sample multiple individuals carrying alleles of moderate frequency (5% or higher). Written informed consent was obtained from each subject and/or their parent as part of a protocol approved by the Stanford University Institutional Review Board (#40479). To help control for weekly batch effects, subject 1111 donated fresh blood for each parasite assay. Eleven other subjects donated blood on at least 2 different weeks, constituting biological replicates. Whole blood samples from a HE patient were obtained from Dr. Bertil Glader under a separate approved protocol (Stanford IRB #14004) that permitted sample sharing among researchers. All samples were de-identified upon collection by labeling with a random four-to-six digit code. Two samples were eventually removed from analysis based on a failed sequencing library (6449KD) and history of stem cell transplant (8715).

Whole blood was drawn into CPDA tubes and spun down within 36 hr to separate serum, buffy coat, and RBCs. RBCs were washed and stored in RPMI-1640 medium (Sigma-Aldrich) supplemented with 25 mM HEPES, 50 mg/L hypoxanthine, and 2.42 mM sodium bicarbonate at 4°C. Buffy coat was transferred directly to cryotubes and stored at –80°C.

## Exome sequencing and genotype calling

Genomic DNA was isolated from frozen buffy coats using a DNeasy Blood and Tissue Kit (QIAGEN). Libraries were prepared using a KAPA Hyperplus Kit (Roche) and hybridized to human exome probes using the SeqCap EZ Prime Exome Kit (Roche). The resulting exome libraries were sequenced with paired-end 150 bp Illumina reads on the HiSeq or NextSeq platforms at Admera Health (South Plainfield, NJ).

Reads were aligned to the hg38 human reference genome using bwa mem (*Li, 2013a*), yielding an average coverage of 42 X across targeted exome regions (excluding sample 6449KD). Variants were called using GATK best practices (*Van der Auwera et al., 2013*) and hard filtered with the following parameters: QD<2.0, FS>60.0, ReadPosRankSum<–2.5, SOR>2.5, MQ<55.0, MQRankSum<–1.0, and DP<500. To minimize the effects of sequencing errors, variants not present in 1000 Genomes, dbSNP_138, or the Mills indel collection (*Mills et al., 2006*) were discarded. Variants that were significantly more frequent in our sample than in gnomAD African and European populations (*Karczewski et al., 2020*) were also discarded, in order to avoid false associations from miscalled variants. We also excluded singleton variants from all association analyses, potentially including some variants unique to other populations. With the remaining variants, we calculated kinship coefficients among all pairs of donors using vcftools `--relatedness2`. Only the six members of the known family had pairwise coefficients >0.044, confirming that no other donors were related.

PIEZO1 E756del was genotyped via PCR and Sanger sequencing according to a previously published protocol (*Nguetse et al., 2020*). To call deletion variants that cause α-thalassemia in the paralogous genes HBA2 and HBA1, we extracted reads from each .bam file that lacked any mismatches or soft-clipping and had MAPQ≥13 (i.e., <5% chance of mapping error). Coverage with these well-mapped reads was calculated over the 73 and 81 bp of unique sequence in HBA2 and HBA1 and normalized to each sample's exome-wide coverage. To determine which samples has unusually low coverage, we formed an ad hoc reference panel of seven donors who were unlikely to carry deletion alleles based on their normal MCH, MCV, and HGB and >96% exome-wide European ancestry (*Weatherall, 2001*). We called heterozygous HBA2 deletions when normalized coverage across three unique regions of the HBA2 gene was below the minimum reference value. Similarly, we called homozygous HBA2 deletions when normalized coverage across three unique regions of the HBA2 gene was less than half of the minimum reference value. This approach resulted in an estimated HBA2 copy number of 2.0 in the reference panel, 0.95 in eight putative heterozygotes and 0.12 in four putative homozygotes. The same method produced no evidence of HBA1 deletion in any sample.

## Variant classification and linkage pruning

Exonic variants in RefSeq genes were identified using ANNOVAR (*Wang et al., 2010*). Variants were classified into three categories: those within 23 malaria-related genes (*Figure 5—source data 1*); those within 887 other RBC proteins (*Figure 5—source data 2*) derived with a medium-confidence filter from the Red Blood Cell Collection database (rbcc.hegelab.org); and those within any other gene.

Linkage between all pairs of bi-allelic, exonic variants in our 121 genotyped samples was calculated using the `--geno-r2 and --interchrom-geno-r2` functions in vcftools (*Danecek et al., 2011*). Variants in RBC genes that shared $r^2$ >0.1 with any variant in the 23-gene set were removed. Within the 23-gene set and RBC-gene set separately, non-carrier variants were ranked by the p-values of their OLS regression with all four parasite measures. Then, one variant was removed from each pair with $r^2$ >0.1, prioritizing retention in the following order: greater significance across models; non-synonymous protein change; higher frequency in our sample; and finally by random sampling. We report results from additive genetic models (genotypes coded 0/1/2), which performed as well or better than recessive (0/0/2) and dominant (0/2/2) models. For three variants, overdominant models (0/1/–) provided the best fit and were used to estimate effect sizes (*Figure 5—figure supplement 3*).

## Population analysis

The population ancestry of our donors was assessed by comparison with African, European, East Asian, and South Asian reference populations from the 1000 Genomes Project (*Auton et al., 2015*). Briefly, variants called from an hg38 alignment of the 1000 Genomes data (*Lowy-Gallego et al., 2019*) were filtered for concordance with the variants genotyped in this study. The `--indep-pairwise`

command in PLINK (*Purcell et al., 2007*) was used to prune SNPs with $r^2$ >0.1 with any other SNP in a 50-SNP sliding window, producing 35,759 unlinked variants. These variants were analyzed in both PLINK --pca and in ADMIXTURE (*Alexander and Lange, 2011*) with K=4 for the 121 genotyped individuals in this study, alongside 2458 individuals from 1000 Genomes. Pan-African and pan-European allele frequencies were obtained from gnomAD v3 (*Karczewski et al., 2020*). $F_{ST}$ for specific alleles was calculated as $(H_T - H_S)/H_T$ and then polarized, such that positive values indicate variants more common in Africa.

## *P. falciparum* culture and assays

Our 3D7 strain of *P. falciparum* was obtained from the Walter and Eliza Hall Institute (Melbourne, Australia) and routinely cultured in human erythrocytes obtained from the Stanford Blood Center. Th.026.09 is a clinical strain isolated from a patient in Senegal in 2009 and kindly provided by Daouda Ndiaye and Sarah Volkman. 3D7 is drug-sensitive and has been lab-adapted for over 40 years, whereas Th.026.09 is drug-resistant and minimally lab-adapted (*Walliker et al., 1987*; *Daniels et al., 2012*; *Moser, 2020*). 3D7 was maintained at 2% hematocrit in RPMI-1640 supplemented with 25 mM HEPES, 50 mg/L hypoxanthine, 2.42 mM sodium bicarbonate, and 4.31 mg/ml Albumax (Invitrogen) at 37°C in 5% $CO_2$ and 1% $O_2$. Th.026.09 was maintained in the same conditions, except that half the Albumax was replaced with heat-inactivated human AB serum.

Parasite growth and invasion assays were performed using schizont-stage parasites isolated from routine culture using a MACS magnet (Miltenyi). Parasites were added at ~0.5% initial parasitemia to fresh erythrocytes suspended at 1% hematocrit in complete RPMI, as above. Parasites were cultured in each erythrocyte sample for 3–5 days in triplicate 100 µl wells. Parasitemia was determined as the average of the three technical replicates, excluding single outlier points, on day 0, day 1 (24 hr), day 3 (72 hr), and in some cases day 5 (120 hr). The fraction of infected RBCs was measured by staining with SYBR Green one nucleic acid stain (Invitrogen, Thermo Fisher Scientific, Eugene, OR) at 1:2000 dilution in PBS/0.3% BSA for 20 min, followed by flow cytometry analysis on a MACSQuant flow cytometer (Miltenyi). Raw invasion rate was defined as the day 1 parasitemia divided by the day 0 parasitemia; raw growth rate was defined as the day 3 (or day 5) parasitemia divided by the day 1 (or day 3) parasitemia. Day 0 parasitemia was not measured in weeks 1–3, so invasion rate estimates are absent for these samples (N=58 unrelated non-carriers with invasion data). The parasite assays failed for both strains in week 9 and for Th.026.09 in week 10, and so were repeated in weeks 10 and 11 with RBCs that had been stored for 1 or 2 weeks.

To correct for batch effects, including substantial week-to-week variation in *P. falciparum* replication rate, we extracted the residuals from a linear regression of the raw parasite values against up to four significantly related batch variables: (1) the raw values for control donor 1111 each week; (2) the parasitemia measured at the previous time point; (3) the age in weeks of the RBCs being measured; and (4) the experimenter performing the assays. Notably, there was no additional effect of 'Week' or the length of the experiment (i.e., 3 or 5 days) once the above variables were regressed out. To convert these residuals (mean 0%) to relative percentages (mean 100%), we first trained linear models for growth and invasion in each strain with data from control donor 1111 and carriers with extreme parasite values (HbAS and HE for growth; G6PD$^-_{high}$ and HE for invasion). For these models, relative percentages were calculated by normalizing the raw multiplication rates in these samples to the raw multiplication rate in the 1111 control from that week. These linear models were used to convert residuals to relative percentages for all samples. Finally, the relative percentages were arithmetically adjusted so that the mean invasion and growth values for non-carriers was 100%. Code for this normalization is available at https://github.com/emily-ebel/RBC (copy archived at swh:1:rev:31f953428a4ec5f0fa83201085ada0a0995facb2), *Ebel, 2021*.

## Red cell phenotyping and normalization

Complete blood count (CBC) data for RBCs, reticulocytes, and platelets were obtained with an ADVIA 120 hematology analyzer (Siemens, Laguna Hills, CA) at the Red Cell Laboratory at Children's Hospital Oakland Research Institute. These data were: RBC, HGB, HCT, MCV, MCH, MCHC, CHCM, RDW, HDW, PLT, MPV, Reticulocyte number and percentage, and the fraction of RBCs in each of nine cells of the RBC matrix (see *Figure 3—figure supplement 2*). Systematic biases were evident for some

measures in certain weeks, but data from control donor 1111 were not available for all weeks. Therefore, CBC data were normalized such that the median value for non-carrier samples was equal across weeks.

Osmotic fragility tests were performed in duplicate by incubating 20 µl of washed erythrocytes for 5 min in 500 µl solutions of NaCl in 14 concentrations: 7.17, 6.14, 5.73, 5.32, 4.91, 4.50, 4.30, 4.09, 3.89, 3.68, 3.27, 3.07, 2.66, and 2.46 g/L. Tubes were spun for 5 min at 1000 g and 100 µl of supernatant was transferred to a 96-well plate. Hemoglobin concentration was determined by adding 100 µl of Drabkin's reagent (Ricca Chemical) to each well and measuring absorbance at $OD_{540nm}$ with a Synergy H1 Plate Reader (Biotek). Relative lysis was determined by normalizing to the maximum hemoglobin concentration in the 14-tube series for each sample. After outlier points were manually removed, sigmoidal osmotic fragility curves were estimated under a self-starting logistic model in the nls package in R. Curves were summarized by the relative tonicity at which 50% lysis occurred (see *Figure 3—figure supplement 4*) and normalized within weekly batches, such that this value was equal for control sample 1111 across weeks.

Osmotic gradient ektacytometry (*Clark et al., 1983*; *Kuypers, 1990*) was performed at the Red Cell Laboratory at Children's Hospital Oakland Research Institute. Red cell deformability estimates across a gradient of NaCl concentrations were fitted to a 20-parameter polynomial model to generate a smooth curve, which was manually verified to closely fit the data. Each curve was summarized with three standard points (*Figure 3—figure supplement 5*; *Clark et al., 1983*), which were normalized such that the median x- and y-values of the three points was equal for non-carrier samples across weeks.

## Statistical analysis

Student's t-test was used to compare trait values between non-carriers and carriers where N>1. Given our modest sample sizes and the expected noise in parasite data, we defined statistical significance as p<0.1. Where N=1 (i.e., for $G6PD^-_{high}$ and HE), significance was assessed with the percentile of the non-carrier distribution. For all comparisons of two continuous variables, OLS linear regression was performed with the lm function in R unless otherwise specified. Adjusted $R^2$ values are reported.

LASSO regression (*Tibshirani, 1994*; *Chatterjee, 2013*) was performed in a k-folds CV framework with the glmnet and caret packages in R. For each of 1000 iterations, we used the createFolds function with k=10 to split the non-carrier data into 10-folds of roughly equal size. Each fold was used as a 'test set' for a LASSO model trained on the remaining nine folds. For each of the 1000 iterations in which 10-folds were created, we collected 10 sets of predictors from the 10 train sets; one average $R^2$ value for the 10 train sets; and one average $R^2$ for the 10 test sets. Each set of 1000 resulting $R^2$ values were normally distributed, and their average is reported in *Figures 4 and 5*. The fraction of k-folds CV support per predictor is based on 10,000 total train models (1000 iterations*10 folds each) and is reported in *Figure 4A* and *Figure 5—source data 3*.

To perform LASSO with each training set, we used the cv.glmnet function with α=1. This function split the train data into 10 folds 11 times, the first to estimate a lambda sequence and the rest to compute the fit with one fold omitted. The lambda value that produced minimal error in the training data was then used to predict values in the independent test data described above. Since cv.glmnet selects folds at random, we performed this procedure five times for each train/test set (which we term 'internal cross-validation'). We retained $R^2$ values and selected predictors from the median model of these five internal CVs. Internal CVs did not otherwise contribute to the k-folds CV support reported in the main text.

To assess the significance of each LASSO result, we applied the same modeling procedure to 1000 data sets with randomly permuted parasite values, which preserved the original correlations among RBC predictors. We performed 10 iterations of fold creation for each permuted data set and retained the average $R^2$ for each set of 10-folds, which generated 1000 fold-averaged $R^2$ values for train sets and 1000 fold-averaged $R^2$ values for train sets. Significance was determined by the percentile of the permuted distribution in which the real data fell. We also applied this same procedure to 1000 sets of 23 genes chosen at random from the RBC proteome (*Figure 5—source data 2*).

We noticed that LASSO effect size estimates for each predictor varied considerably across models. Therefore, we used univariate OLS regression on all non-carrier data (excluding five of the six family members) to estimate the effect size of each predictor selected at least once by LASSO. OLS p-values

are reported as a measure of confidence in these effect size estimates, with $p<0.1$ considered sufficient evidence to report the effect size. However, because OLS regression was only performed for variants pre-selected by LASSO, these p-values cannot be interpreted on their own as evidence of significant associations.

We compared groups of selected genetic variants using the binom.test function in R. For synonymous alleles, we used the proportion of synonymous alleles in the input set of 106 variants (53%) as the null hypothesis. For allele frequencies in Africa and Europe, we categorized protective variants as more common (to any absolute degree) among Africans (N=21,042) or non-Finnish Europeans (N=32,399) in the gnomAD database. The null hypothesis was that 50% of the alleles would be more common in Africans.

## Acknowledgements

The authors gratefully acknowledge the invaluable participation of all volunteer blood donors. Nick Bondy, Bertil Glader, Sandra Larkin, Brian Fleischer, Ashley Dunn, Talal Seddik, Trung Pham, David Vu, and Spectrum Child Health provided crucial assistance in donor coordination and sample processing. *P. falciparum* strain Th.026.09 was kindly provided by Daouda Ndiaye and Sarah Volkman. For quantitative advice, the authors thank Grant Kinsler, Jonathan Pritchard, Susan Holmes, and the Stanford Statistics Consulting Group. This study was primarily supported by a Pilot Early Career award from the Stanford Maternal Child Health Research Institute and a Gabilan Faculty Award from the Stanford University School of Medicine Office of Faculty Development and Diversity (ESE). ERE was an NSF Graduate Research Fellow (DGE-1247312) and received additional support from the Stanford Center for Computational, Evolutionary, and Human Genomics. DAP was funded through an NIH MIRA award 5R35GM118165-05. ESE is a Tashia and John Morgridge Endowed Faculty Scholar in Pediatric Translational Medicine through the Stanford Maternal Child Health Research Institute. Local blood samples were drawn at the Stanford Clinical and Translational Research Unit, which is supported by CTSA Grant UL1 TR001085.

## Additional information

### Funding

| Funder | Grant reference number | Author |
| --- | --- | --- |
| Stanford University School of Medicine | | Elizabeth S Egan |
| Stanford University School of Medicine | | Elizabeth S Egan |
| Stanford Center for Computational, Evolutionary and Human Genomics | | Emily R Ebel |
| National Institute of General Medical Sciences | 5R35GM118165-05 | Dmitri A Petrov |
| National Science Foundation | DGE-1247312 | Emily R Ebel |

The funders had no role in study design, data collection and interpretation, or the decision to submit the work for publication.

### Author contributions

Emily R Ebel, Conceptualization, Formal analysis, Funding acquisition, Investigation, Methodology, Visualization, Writing – original draft, Writing – review and editing; Frans A Kuypers, Investigation, Methodology, Resources, Writing – review and editing; Carrie Lin, Investigation; Dmitri A Petrov, Conceptualization, Methodology, Resources, Supervision, Writing – review and editing; Elizabeth S Egan, Conceptualization, Investigation, Methodology, Resources, Supervision, Writing – review and editing

## Author ORCIDs
Emily R Ebel (iD) http://orcid.org/0000-0002-3244-4250
Dmitri A Petrov (iD) http://orcid.org/0000-0002-3664-9130
Elizabeth S Egan (iD) http://orcid.org/0000-0002-2112-7700

## Ethics

Human subjects: Written informed consent and consent to publish was obtained from each subject and/or their parent as part of a protocol approved by the Stanford University Institutional Review Board (#40479).

## Decision letter and Author response

Decision letter https://doi.org/10.7554/eLife.69808.sa1
Author response https://doi.org/10.7554/eLife.69808.sa2

---

# Additional files

## Supplementary files
• Transparent reporting form

## Data availability

All data generated or analyzed during this study are included in the manuscript and supporting files. Source data files have been provided for Figures 1, 4, and 5 and other raw data and normalization scripts are available at https://github.com/emily-ebel/RBC (copy archived at https://archive.software-heritage.org/swh:1:rev:31f953428a4ec5f0fa83201085ada0a0995facb2).

The following dataset was generated:

| Author(s) | Year | Dataset title | Dataset URL | Database and Identifier |
|---|---|---|---|---|
| Ebel ER, Kuypers FA, Lin C, Petrov DA, Egan ES | 2020 | Exome Sequencing from Participants in RBC/Malaria Study | https://www.ncbi.nlm.nih.gov/bioproject/PRJNA683732/ | NCBI BioProject, PRJNA683732 |

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
