## [Decision Letter]

**Acceptance summary:**

This paper finds that common red blood cell phenotypic and genetic variation predicts susceptibility to malarial parasites. Contrary to hypotheses about ancestry-associated malaria selection, however, these variants are not more common in African ancestry populations. Overall, this work presents convincing evidence that in vitro assays of malarial invasion and growth are a practical, effective complement to large-scale genome-wide association studies for understanding the genetics of malarial infection.

**Decision letter after peer review:**

Thank you for submitting your article "Common host variation drives malaria parasite fitness in healthy human red cells" for consideration by *eLife*. Your article has been reviewed by 3 peer reviewers, one of whom is a member of our Board of Reviewing Editors, and the evaluation has been overseen by George Perry as the Senior Editor. The reviewers have opted to remain anonymous.

Essential revisions:

1. Demonstrate the robustness of the results to removing the children from the single large family (one mother and five children) currently included in the analysis. More generally, demonstrate that genotype-based prediction is not confounded with family membership/relatedness.

2. Demonstrate that the LASSO model retains high accuracy when predicting Plasmodium invasion and growth phenotypes out-of-sample. Here, it will be crucial to completely separate the training set for the model from the test set to which it is applied (beyond internal cross-validation), either by cleanly stratifying the current sample or ideally by extending to new samples.

3. Evaluate the invasion measurements at 72 hours (the "re-invasion" phenotype); consider whether this reduces the very large amount of noise associated with the original 24-hour invasion phenotype.

4. Discuss the generalizability of the current findings to strains beyond the two strains (one lab, one clinical isolate) used in this study, including the rationale for the choice of these strains and the differences in results between them.

*Reviewer #1 (Recommendations for the authors):*

1. I'm most concerned about the estimates of predictive accuracy/generalization error. The permutations used to assess predictive accuracy confirm that the particular set of variants chosen by LASSO are more predictive than randomly chosen variants, in this particular sample. However, they don't provide insight into the predictive accuracy of the model out of sample. Although cross-validation should help with that problem, the CV procedure used in glmnet was not clear (also, I assume that α was fixed to 1 throughout; i.e., the authors only used LASSO, not the elastic net-it would be helpful to provide the exact parameters used). As reflected in my public review, I'm also surprised to see such strong predictive accuracy when the repeatability of growth and invasion measures from the same individuals sampled in different weeks (Figure S1) is modest to low. Is the repeatability much higher after controlling for batch and technical effects (which appear to be very substantial based on Table S1)?

2. Related, the predictive accuracy is so good that the results and methods would be very compelling if the model truly generalizes. Towards that end, I think it is essential to use a true out-of-sample test set. Ideally, this could be done by collecting additional samples and phenotyping/genotyping them. Minimally, a cleaner training/test split could be accomplished, e.g., by fitting the model with n = 50 (using internal CV) and then predicting out of sample in the remaining n = 23-although this compromises sample size in the training set, the model prediction accuracy is so high that it should be robust (note that if this approach is used, it will be important not to leak information from the training set into the test set during data normalization-that is, the values from the training set should not be allowed to influence the values from the test set at all). An additional approach would be to predict the repeated sample phenotype values (from n = 11 donors) based on the n = 73 non-carrier donors with one sample represented per donor (not as good, because the samples in the test set would not be truly independent, but still instructive).

3. If the predictive accuracy does hold up, I think the remarkably large effect sizes need to be reconciled with the difficulty of identifying large effect hits in malaria GWAS. Is this expected based on the strength of the correlation between replication rates in vivo and malaria infection/progression? Are the variants identified in the LASSO model strongly enriched for low p-values in GWAS (beyond linkage to known hits for some subset of variants)?

*Reviewer #2 (Recommendations for the authors):*

Awesome paper! It was a pleasure to read and very well written. The attention to detail was greatly appreciated. Most of my private recommendations are mainly suggestions for how to improve the presentation of data, but none of them are vital to the manuscript.

1. This isn't necessary, but I would like to suggest a figure that shows the association (pairwise) by carrier status for all of the RBC traits and invasion/growth rate statuses. This could be a heatmap where you would be able to show that certain carriers have a certain pattern of outcomes. You have this already in the text, but it may be easier for the reader to see it in figure format.

2. Most of my private recs are just about figures. Would it be possible to also include the association of RBC traits and African ancestry in Figure 6? I think these are really interesting and not having them in Figure 6 undersells the findings.

3. The scatterplots with the transparent dots are a little confusing to see. I would suggest something like a beeswarm plot for plots like Figure 2A-B, with a separate column for the replicates to show the tight distribution.

4. The first sentence of the discussion reads that "healthy red blood cells (RBCs) harbor extensive phenotypic and genetic variation,". RBCs have no nucleus and therefore no DNA.

*Reviewer #3 (Recommendations for the authors):*

1. The invasion measurements (fold change parasitaemia over 24 hours) were subject to a tremendous amount of variation, perhaps owing to culture conditions affecting schizont egress and subsequent merozoite invasion of RBCs. The authors acknowledged that these environmental effects could lead to greater experimental noise. How would their invasion measurements and analyses change if they took the parasitaemia measurements of parasites that had already gone through one life-cycle in the test RBCs, e.g. at 72 hours (re-invasion measurements)?

2. Limitations in targeted gene approach: could there be non-identified "disease alleles" in non-carriers that explain the overlap in RBC phenotypes and parasite fitness with carriers? They categorised carriers as those with known RBC disease alleles, mainly in haemoglobin and G6PD genes, while non-carriers as those not carrying these alleles. The genetic variants that they added to their analysis were limited to membrane protein genes. The non-carriers could carry a spectrum of additional gene variants that impact the RBC phenotypes observed, which could therefore influence parasite fitness.

3. The authors used one lab parasite strain and one field parasite isolate for their study, wouldn't it have been beneficial to also select a variety of parasite strains representing different invasion pathways and growth patterns, to check if these genetic and RBC phenotypic factors hold true across different strains? Given the limitations with the field isolate, wouldn't it be worthwhile to test other lab strains that use alternative invasion pathways? Also, it would be good to provide a sentence or two explaining the choice of lab and field strains in the study.

4. It was surprising that no variants in the glycophorins and haemoglobin genes were detected, given their important roles in the function of the RBC, and in parasite invasion (in the case of the glycophorins). They have previously been found to have large effect sizes in populations living in malaria endemic regions. Could the authors discuss this?

[Editors' note: further revisions were suggested prior to acceptance, as described below.]

Thank you for resubmitting your work entitled "Common host variation drives malaria parasite fitness in healthy human red cells" for further consideration by *eLife*. Your revised article has been evaluated by George Perry as the Senior Editor, and a Reviewing Editor.

The manuscript and response to the previous reviews address nearly all the original reviewer comments and concerns and, overall, represent an excellent contribution to the literature. Revisions to the LASSO prediction analysis now present convincing and realistic evidence that red blood cell phenotypes and common RBC alleles help predict in vitro growth phenotypes.

The remaining issue to be addressed is the inclusion of statistics on training set variance explained as a major result in the text, and as key parts of Figures 4 and 5 (parts B and C of each figure). As the reduction in explanatory power in the external test sets shows, these estimates are over-optimistic and likely a result of overfitting. We ask that you remove the training set statistics from the results and figures, as the test set results alone provide a clearer, more accurate view of model performance to readers, and likely mitigate concerns from readers who are experienced with (and concerned about) overfitting in predictive modeling.

---

## [Author Response]

Essential revisions:1. Demonstrate the robustness of the results to removing the children from the single large family (one mother and five children) currently included in the analysis. More generally, demonstrate that genotype-based prediction is not confounded with family membership/relatedness.

We have repeated all the statistical analyses and updated the results after excluding the five children. The major genotypic and phenotypic predictors of *P. falciparum* replication remain basically the same, except for the family-specific PCs. We interpret this to mean that population structure from the family is unlikely to have biased the results, although some features of those samples are clearly associated with parasite fitness. We now note throughout the manuscript that the children are omitted from the figures and association analyses. To further protect from family confounding, we have also calculated pairwise kinship coefficients using ~400,000 SNPs (Methods) to confirm that no other related individuals were present in our data set.

2. Demonstrate that the LASSO model retains high accuracy when predicting Plasmodium invasion and growth phenotypes out-of-sample. Here, it will be crucial to completely separate the training set for the model from the test set to which it is applied (beyond internal cross-validation), either by cleanly stratifying the current sample or ideally by extending to new samples.

As suggested, we have performed k-fold cross validation on training and test sets to demonstrate that the LASSO model retains high accuracy. We have substantially revised the code, Methods, Results, and two main figures to reflect the suggested re-analysis. Over 1000 random splits of the non-carrier data into 10 folds, we found that RBC traits/genotypes selected from ‘train’ data have prediction accuracy in the separate ‘test’ fold that is significantly higher than expected from shuffled data or random genes. Many of the specific RBC predictors remain the same as in the earlier analysis, although the absolute variance explained in the test data is much smaller than in the train data (max R^2^ ~ 15% with N ~ 7 vs. ~80% with N ~60). We agree that this true cross-validation approach helps avoid overfitting while still identifying important RBC predictors of *P. falciparum* growth rate. Overall, the fact that our model retains significant predictive power in out-of-sample data provides strong support for our experimental approach and will stimulate additional research in this area using new and larger groups of samples.

3. Evaluate the invasion measurements at 72 hours (the "re-invasion" phenotype); consider whether this reduces the very large amount of noise associated with the original 24-hour invasion phenotype.

We agree that the manuscript would be improved if the measurements of *P. falciparum* invasion (24 hour timepoint) were less noisy. In line with the reviewer’s suggestion, we measured the parasitemia at 72 hours (the “re-invasion” phenotype), though please note that this was referred to as “growth” in the original manuscript because it reflects the growth of a ring stage parasite through a complete life cycle of development, egress, and re-invasion. We have altered our experimental diagram in Figure 1C to more clearly indicate that “re-invasion” is part of our “growth” measurement. Notably, we found genetic variants in well-known invasion receptors to be associated only with growth (Figure 5A). We discuss that several RBC phenotypes are significantly associated with both invasion and growth (Figure 4A), although invasion is noisier for both technical and biological reasons. The discussion now acknowledges our limitations in measuring invasion and offers suggestions for future experiments (lines 551-564).

4. Discuss the generalizability of the current findings to strains beyond the two strains (one lab, one clinical isolate) used in this study, including the rationale for the choice of these strains and the differences in results between them.

We have included more details and citations for the two divergent strains in the results and methods (Lines 146-150, 747-749). We now discuss the strong correlations between the strains, including for specific phenotypes and genotypes, which suggest that our results may be generalizable (Lines 565-575). Given some interesting differences between the strains, we acknowledge that future work would benefit from assaying additional parasite diversity linked to specific invasion pathways.

Reviewer #1 (Recommendations for the authors):1. I'm most concerned about the estimates of predictive accuracy/generalization error. The permutations used to assess predictive accuracy confirm that the particular set of variants chosen by LASSO are more predictive than randomly chosen variants, in this particular sample. However, they don't provide insight into the predictive accuracy of the model out of sample. Although cross-validation should help with that problem, the CV procedure used in glmnet was not clear (also, I assume that α was fixed to 1 throughout; i.e., the authors only used LASSO, not the elastic net-it would be helpful to provide the exact parameters used). As reflected in my public review, I'm also surprised to see such strong predictive accuracy when the repeatability of growth and invasion measures from the same individuals sampled in different weeks (Figure S1) is modest to low. Is the repeatability much higher after controlling for batch and technical effects (which appear to be very substantial based on Table S1)?

We thank the reviewer for this important critique. As suggested, we have performed k-fold cross validation on training and test sets to demonstrate that the LASSO model retains high accuracy out-of-sample. Over 1000 random splits of the non-carrier data into 10 folds, we found that RBC traits/genotypes selected from ‘train’ data have prediction accuracy in separate ‘test’ folds that is significantly higher than expected from shuffled data or random genes. This analysis adds insight into the predictive accuracy of the selected phenotypes and genotypes in true out-of-sample data, which as expected, is lower than in the same data used to train the models. We have updated Figures 4 and 5 and the accompanying Results text (Lines 253-354, 392-413) to reflect the change in analysis. We apologize for the lack of clarity in the Methods, which have been revised to better explain all cross-validation procedures and parameters used (Lines 824-850). We have also revised the discussion (Lines 507-509; 554-559) to clarify that both biological and technical variation are expected to produce noise across samples from the same individuals collected over weeks to months. Finally, we have clarified the caption of Figure 2—figure supplement 2 to indicate that the data are shown after batch correction.

2. Related, the predictive accuracy is so good that the results and methods would be very compelling if the model truly generalizes. Towards that end, I think it is essential to use a true out-of-sample test set. Ideally, this could be done by collecting additional samples and phenotyping/genotyping them. Minimally, a cleaner training/test split could be accomplished, e.g., by fitting the model with n = 50 (using internal CV) and then predicting out of sample in the remaining n = 23-although this compromises sample size in the training set, the model prediction accuracy is so high that it should be robust (note that if this approach is used, it will be important not to leak information from the training set into the test set during data normalization-that is, the values from the training set should not be allowed to influence the values from the test set at all). An additional approach would be to predict the repeated sample phenotype values (from n = 11 donors) based on the n = 73 non-carrier donors with one sample represented per donor (not as good, because the samples in the test set would not be truly independent, but still instructive).

We appreciate this comment, which inspired a thorough re-analysis of the non-carrier data in a k-folds cross-validation framework. In addition to the previous responses, we here note that model performance on held-out test data was highly dependent on which samples were randomly allocated into the (small) test set. To avoid bias from a single randomization of the data into one test and one train set, we performed 1,000 random divisions of the data into 10 folds and treated each fold in turn as a left-out test set. We now report the mean performance in 10*1,000 = 10,000 test sets and 10,000 train sets, which should be more representative of the predictive power of the selected variables.

With respect to leaking information from the test set to the train set, we agree that it is critical to ensure that the parasite values in each set are not unduly influenced by each other. To that end, the parasite data from non-carriers besides the repeated control (1111) were not used to batch-correct the parasite data of other samples (see Figure 2—figure supplement 1 and the updated description of parasite normalization, Lines 776-785).

Unlike with the parasite data, we performed a median-based normalization of RBC phenotypes only once using all the non-carrier data. This was a different type of normalization based on equalizing weekly medians, which we chose because we lacked weekly control data (from sample 1111) for about half the weeks for most RBC phenotypes. If this phenotype normalization produced significant leakage across test and train sets, we might expect a spurious signal in models combining phenotypes with random genes. Importantly, we do not detect this (Figure 5), suggesting any leakage from phenotype normalization is minor.

Finally, we have added text to clarify that RBC phenotypes from the same individual are expected to vary over time (Lines 507-509). Unfortunately, we were unable to directly test our models on the repeated samples because we only collected parasite data (without other RBC phenotypes) for repeated samples after the first assay. We agree that intra-sample variability is an interesting avenue for future, more comprehensive research.

Overall, we appreciate this collection of important and thoughtful recommendations. They motivated a re-analysis that strongly increases our confidence in the RBC phenotypes and genotypes associated with *P. falciparum* replication and strengthens the manuscript.

3. If the predictive accuracy does hold up, I think the remarkably large effect sizes need to be reconciled with the difficulty of identifying large effect hits in malaria GWAS. Is this expected based on the strength of the correlation between replication rates in vivo and malaria infection/progression? Are the variants identified in the LASSO model strongly enriched for low p-values in GWAS (beyond linkage to known hits for some subset of variants)?

RBCs are certainly important for malaria, and we agree that future studies should take further advantage of existing GWAS data. Nevertheless, we would not expect large RBC effects on parasites in our experiments to translate to large effects in severe malaria GWAS (SM-GWAS). This is primarily because individual immune response, shaped by malaria exposure and age, is a major factor determining SM risk. Interestingly, the main new locus discovered by SM-GWAS (which changes expression of the RBC ion channel *ATP2B4*) has been associated with both immune cell and RBC phenotypes. A disconnect in *ATP2B4* effect sizes for SM-GWAS and RBC traits has also been noted in recent work (Band et al., 2019; Villegas-Mendez et al., 2021). The present association study is the first of its kind in RBCs, and we are reassured that more than half of our growth-associated variants have already been associated with other RBC traits by GWAS, suggesting they are functional or linked to functional variants.

Reviewer #2 (Recommendations for the authors):Awesome paper! It was a pleasure to read and very well written. The attention to detail was greatly appreciated. Most of my private recommendations are mainly suggestions for how to improve the presentation of data, but none of them are vital to the manuscript.1. This isn't necessary, but I would like to suggest a figure that shows the association (pairwise) by carrier status for all of the RBC traits and invasion/growth rate statuses. This could be a heatmap where you would be able to show that certain carriers have a certain pattern of outcomes. You have this already in the text, but it may be easier for the reader to see it in figure format.

Thank you for this suggestion. We have added a heatmap showing the phenotypic patterns for each carrier group as a supplement for Figure 3.

2. Most of my private recs are just about figures. Would it be possible to also include the association of RBC traits and African ancestry in Figure 6? I think these are really interesting and not having them in Figure 6 undersells the findings.

We agree that the relationship between RBC and African ancestry is very interesting. We have added three of these panels to Figure 6, with the rest available in the figure supplement.

3. The scatterplots with the transparent dots are a little confusing to see. I would suggest something like a beeswarm plot for plots like Figure 2A-B, with a separate column for the replicates to show the tight distribution.

Thank you for this suggestion. We agree that beeswarm plots are very effective for comparing two groups, such as in the new Figure 5—figure supplement 2 and Figure 5—figure supplement 5. When comparing multiple groups, such as in Figures 2 and 3, we found beeswarm plots awkwardly wide when disallowing overlapping points. We now show similar points overlapping at the edges of each column (option “gutter”), which we agree offers a more accurate representation of the distributions than the transparent points.

4. The first sentence of the discussion reads that "healthy red blood cells (RBCs) harbor extensive phenotypic and genetic variation,". RBCs have no nucleus and therefore no DNA.

We have changed “genetic” to “proteomic” in this sentence to avoid confusion.

Reviewer #3 (Recommendations for the authors):1. The invasion measurements (fold change parasitaemia over 24 hours) were subject to a tremendous amount of variation, perhaps owing to culture conditions affecting schizont egress and subsequent merozoite invasion of RBCs. The authors acknowledged that these environmental effects could lead to greater experimental noise. How would their invasion measurements and analyses change if they took the parasitaemia measurements of parasites that had already gone through one life-cycle in the test RBCs, e.g. at 72 hours (re-invasion measurements)?

We agree that the manuscript would be improved if the measurements of *P. falciparum* invasion (24 hour timepoint) were less noisy. In line with the reviewer’s suggestion, we measured the parasitemia at 72 hours (the “re-invasion” phenotype), though we referred to this as “growth” because it reflects the growth of a ring stage parasite through a complete life cycle of development, egress, and re-invasion. We have altered our experimental diagram in Figure 1C to more clearly indicate that “re-invasion” is part of our “growth” measurement.

We agree in the discussion (Lines 551-564) that re-invasion might be less noisy than invasion for many reasons, both experimental (e.g. recent time out of the incubator) and biological (e.g. invading host cells of a new genotype). New text in the discussion acknowledges our limitations in measuring invasion and offers suggestions for future experiments.

2. Limitations in targeted gene approach: could there be non-identified "disease alleles" in non-carriers that explain the overlap in RBC phenotypes and parasite fitness with carriers? They categorised carriers as those with known RBC disease alleles, mainly in haemoglobin and G6PD genes, while non-carriers as those not carrying these alleles. The genetic variants that they added to their analysis were limited to membrane protein genes. The non-carriers could carry a spectrum of additional gene variants that impact the RBC phenotypes observed, which could therefore influence parasite fitness.

We agree that unknown ‘disease alleles’ could contribute to the overlap between non-carriers and carriers (Lines 503-507). We analyzed 23 RBC proteins with strong links to *P. falciparum* in the literature, of which 5 were not membrane proteins *(HBB, HBA1, HBA2, G6PD*, and *HP*). We agree that other RBC proteins besides these 23 are likely to contain genetic variation that impacts parasite fitness, although we believe that testing many other, previously unknown proteins would require a much larger sample size (Lines 534-544).

3. The authors used one lab parasite strain and one field parasite isolate for their study, wouldn't it have been beneficial to also select a variety of parasite strains representing different invasion pathways and growth patterns, to check if these genetic and RBC phenotypic factors hold true across different strains? Given the limitations with the field isolate, wouldn't it be worthwhile to test other lab strains that use alternative invasion pathways? Also, it would be good to provide a sentence or two explaining the choice of lab and field strains in the study.

We agree that repeating these experiments with additional strains is a worthwhile endeavor. The current study was primarily limited by our ability to simultaneously culture multiple strains in a limited supply of fresh donor RBCs. For the two divergent strains used, we have included additional details and citations in the results and methods (Lines 146-150, 747-749). We also discuss the strong correlations between the strains, including for specific geneotypes and phenotypes, that suggest that our results may be generalizable (Lines 565-575). Given the interesting differences between the strains in their preference for RBCs with for African ancestry, we acknowledge that future work would benefit from assaying additional parasite diversity linked to specific invasion pathways.

4. It was surprising that no variants in the glycophorins and haemoglobin genes were detected, given their important roles in the function of the RBC, and in parasite invasion (in the case of the glycophorins). They have previously been found to have large effect sizes in populations living in malaria endemic regions. Could the authors discuss this?

In the hemoglobin genes, we did detect the variants that cause α-thalassemia, sickle cell disease, and hemoglobin C disease (Figure 1B). In non-carriers, we found one common, synonymous variant in *HBB* that was rarely selected by LASSO and was not associated with any GWAS data (Figure 5-Source data 3). Only one other common variant in *HBB* exons is present in gnomAD, and though it was not present in our filtered SNP set, it is classified as benign. To clarify the exome variants that we sequenced and analyzed, we have added Figure 1-Source Data 1, which contains annotations and frequencies for all exome variants that passed quality filters.

In the glycophorins, variants with large effect include GYPB*S (Tarazona-Santos et al., 2011), Dantu (Leffler et al., 2017; Field et al., 1994), and deficiencies of GYPA, GYPB, or GYPC. All but GYPB*S are rare in African populations, and we do detect GYPB*S (rs143997559) associated with 3D7 growth. We now point out this specific finding in the results (Lines 402-404).

[Editors' note: further revisions were suggested prior to acceptance, as described below.]

Thank you for resubmitting your work entitled "Common host variation drives malaria parasite fitness in healthy human red cells" for further consideration by eLife. Your revised article has been evaluated by George Perry as the Senior Editor, and a Reviewing Editor.The manuscript and response to the previous reviews address nearly all the original reviewer comments and concerns and, overall, represent an excellent contribution to the literature. Revisions to the LASSO prediction analysis now present convincing and realistic evidence that red blood cell phenotypes and common RBC alleles help predict in vitro growth phenotypes.The remaining issue to be addressed is the inclusion of statistics on training set variance explained as a major result in the text, and as key parts of Figures 4 and 5 (parts B and C of each figure). As the reduction in explanatory power in the external test sets shows, these estimates are over-optimistic and likely a result of overfitting. We ask that you remove the training set statistics from the results and figures, as the test set results alone provide a clearer, more accurate view of model performance to readers, and likely mitigate concerns from readers who are experienced with (and concerned about) overfitting in predictive modeling.

As suggested, we have removed the training set statistics from the results, Figures 4 and 5, and Figure 5 supplementary Figure 1, and now only present the test set statistics.